# Dietary restriction and the transcription factor *clock* delay eye aging to extend lifespan in *Drosophila Melanogaster*

Brian A. Hodge [1✉], Geoffrey T. Meyerhof [1,2], Subhash D. Katewa [1,3], Ting Lian [1,4], Charles Lau [1], Sudipta Bar [1], Nicole Y. Leung [2,5,6], Menglin Li [2], David Li-Kroeger [7], Simon Melov [1], Birgit Schilling [1], Craig Montell [2] & Pankaj Kapahi [1✉]

Many vital processes in the eye are under circadian regulation, and circadian dysfunction has emerged as a potential driver of eye aging. Dietary restriction is one of the most robust lifespan-extending therapies and amplifies circadian rhythms with age. Herein, we demonstrate that dietary restriction extends lifespan in *Drosophila melanogaster* by promoting circadian homeostatic processes that protect the visual system from age- and light-associated damage. Altering the positive limb core molecular clock transcription factor, CLOCK, or CLOCK-output genes, accelerates visual senescence, induces a systemic immune response, and shortens lifespan. Flies subjected to dietary restriction are protected from the lifespan-shortening effects of photoreceptor activation. Inversely, photoreceptor inactivation, achieved via mutating rhodopsin or housing flies in constant darkness, primarily extends the lifespan of flies reared on a high-nutrient diet. Our findings establish the eye as a diet-sensitive modulator of lifespan and indicates that vision is an antagonistically pleiotropic process that contributes to organismal aging.

---

[1] Buck Institute for Research on Aging, 8001 Redwood Blvd, Novato, CA 94945, USA. [2] Neuroscience Research Institute and Department of Molecular, Cellular and Developmental Biology, University of California, Santa Barbara, Santa Barbara, CA 93106, USA. [3] NGM Biopharmaceuticals, 333 Oyster Point Blvd, South San Francisco, CA 94080, USA. [4] Sichuan Agricultural University, 46 Xinkang Rd, Yucheng District, Ya'an, Sichuan, China. [5] Department of Psychiatry and Behavioral Sciences, Stanford University, Stanford, CA 94305, USA. [6] Department of Neurobiology, Stanford University, Stanford, CA 94305, USA. [7] Department of Neurology, Baylor College of Medicine, Houston, TX 77096, USA. ✉email: brianalignbio@gmail.com; pkapahi@buckinstitute.org

Circadian rhythms are ~24 h oscillations in behavior, cellular physiology, and biochemistry, which evolved to anticipate and manage predictable changes associated with the solar day (e.g., predator/prey interactions, nutrient availability, phototoxicity, etc.)[1]. Circadian rhythms are generated by endogenous clocks that sense time-cues (e.g., light and food) to govern rhythmic oscillations of gene transcriptional programs, synchronizing cellular physiology with daily environmental stressors[2]. In addition to keeping time, the molecular clock regulates the temporal expression of downstream genes, known as Clock-controlled genes, to promote tissue-specific rhythms in physiology[3]. The *Drosophila* molecular clock is comprised of transcriptional-translational feedback loops, where the transcription factors Clock (CLK) and Cycle (CYC) rhythmically activate their own repressors, Period (PER) and Timeless (TIM)[2]. This feedback loop not only exists in central pacemaker neurons, where it sets rhythms in locomotor activity, but it also functions in peripheral tissues, such as the eye[4].

Aging is associated with a progressive decline in visual function and an increase in the incidence of ocular disease. *Drosophila* photoreceptor cells serve as a powerful model of both visual senescence and retinal degeneration[5,6]. *Drosophila* and mammalian photoreceptors possess a cell-intrinsic molecular clock mechanism that temporally regulates many physiological processes, including light sensitivity, metabolism, pigment production, and susceptibility to light-mediated damage[7]. Visual senescence is accompanied by a reduced circadian amplitude in core-clock gene expression within the retina[8]. This reduction in retinal circadian rhythms may be causal in eye aging, as mice harboring mutations in their core-clock genes, either throughout their entire body or just in their photoreceptor cells, display several early-onset aging phenotypes within the eye. These mice prematurely form cataracts and have reduced photoreceptor cell light sensitivity and viability[8]. However, the molecular mechanisms by which the molecular clock influences eye aging are not fully understood.

Dietary restriction (DR), defined as reducing specific nutrients or total calories, is the most robust mechanism for delaying disease and extending lifespan[9]. The mechanisms by which DR promotes health and lifespan may be integrally linked with circadian function, as DR enhances the circadian transcriptional output of the molecular clock and preserves circadian function with age[10]. Inversely, high-nutrient diets (i.e., excess consumption of protein, fats, or total calories) repress circadian rhythms and accelerate organismal aging[11,12]. However, how DR modulates circadian rhythms within the eye, and how these rhythms influence DR-mediated lifespan extension, had yet to be examined.

Herein, we sought to elucidate the circadian processes that are activated upon DR by performing an unbiased, 24 h time-course mRNA expression analysis in whole flies. We found that circadian processes within the eye are highly elevated in expression in flies reared on DR. In particular, DR enhanced the rhythmic expression of genes that encode proteins that are involved in light adaptation (i.e., calcium handling and deactivation of rhodopsin-mediated signaling). Building on this observation, we demonstrate that the majority of these circadian phototransduction genes were transcriptionally regulated by CLK. Eliminating CLK function either pan-neuronally, or just in the photoreceptors, accelerated visual decline with age. Furthermore, disrupting photoreceptor homeostasis increased systemic immune responses and shortened lifespan. Several eye-specific CLK-output genes that were upregulated in expression in response to DR, were also required for DR- to slow visual senescence and extend lifespan.

## Results

**Dietary restriction amplifies circadian transcriptional output and delays visual senescence in a CLK-dependent manner.** To determine how DR changes circadian transcriptional output, we performed a series of microarray experiments over the span of 24 h in *Canton-S* wild-type flies (whole fly) housed in light-dark (LD) 12:12 h and reared on either a high-yeast (5%; ad libitum, AL) diet or a low-yeast (0.5%; DR) diet (Supplementary Fig. 1a), identifying rhythmic transcript expression with the circadian algorithm JTK_CYCLE (Supplementary Data 1). A full description of the fly strains utilized in this study is in Supplementary Data 2. In LD conditions, transcripts displaying 24 h oscillations may be driven by the core-molecular clock ("circadian") or by rhythmic lighting cues[13]. To control for the effect of light on rhythmic gene expression, we also performed time-course microarrays under similar conditions in flies that lack an endogenously generated circadian transcriptional rhythm ($tim^{01}$ mutants). We reasoned that the oscillatory transcripts identified in $tim^{01}$ flies are likely driven by light. Therefore, we defined clock-controlled transcripts as those that oscillate only in wild-type flies but not in $tim^{01}$ mutants (Supplementary Fig. 1a). Wild-type flies maintained on DR displayed nearly twice the number of circadian transcripts compared to flies reared on AL (Fig. 1a, b and Supplementary Fig. 1b), while the circadian mutant $tim^{01}$ flies failed to show a significant increase in oscillating transcripts on DR (Supplementary Fig. 1c). These data indicate that the DR-mediated increase in circadian transcripts requires functional molecular clocks and is independent of mechanisms influenced by rhythmic lighting cues (i.e., masking). Circadian gene expression in the wild-type flies was also more robust on DR vs AL; DR-specific oscillators were statistically more rhythmic (lower JTK_CYCLE circadian *P*-values) and displayed larger circadian amplitudes than AL-specific oscillators in the *Canton-S* but not in $tim^{01}$ mutant flies (Supplementary Fig. 1d, e). Diet also drastically altered the wild-type circadian transcriptional profile, as only 16% of DR oscillators were also oscillating on AL (Supplementary Fig. 1b). Furthermore, the AL and DR circadian transcriptomes were enriched for distinct processes (Supplementary Fig. 1f, g and Supplementary Data 1).

Transcripts that oscillate on both AL and DR diets were highly enriched for genes associated with the canonical phototransduction signaling cascade (Fig. 1c, d), which is the process by which *Drosophila* photoreceptor cells, the primary light-sensitive neurons, transduce light information into a chemical signal[14]. Briefly, light-mediated conversion of rhodopsin proteins to their meta-rhodopsin state stimulates heterotrimeric Gq proteins that activate a phospholipase C (NORPA), which produces secondary messengers and promotes the opening of Transient Receptor Potential channels (TRP, TRPL), ultimately allowing $Ca^{2+}$ and $Na^+$ to depolarize the photoreceptor cell[15]. Although the phototransduction transcripts were cyclic on both diets, on DR their expression became more rhythmic (lower JTK_CYCLE *P*-values and larger circadian amplitudes) and elevated (~2-fold increase in expression across all timepoints) (Fig. 1e and Supplementary Fig. 1j). The phototransduction components were similarly upregulated in expression in the $tim^{01}$ mutants, but did not display a 24 h rhythmicity (Supplementary Fig. 1k), indicating that their oscilatory expression patterns are driven by the intrinsic molecular clock, as opposed to being primarily light-responsive. Since our time-course analyses were performed in whole-fly, we queried publicly available circadian transcriptomes from wild-type heads to further investigate the rhythmic oscillations of eye-related transcripts[16]. The majority of the DR-sensitive phototransduction genes also robustly cycled in wild-type heads (Supplementary Table 1).

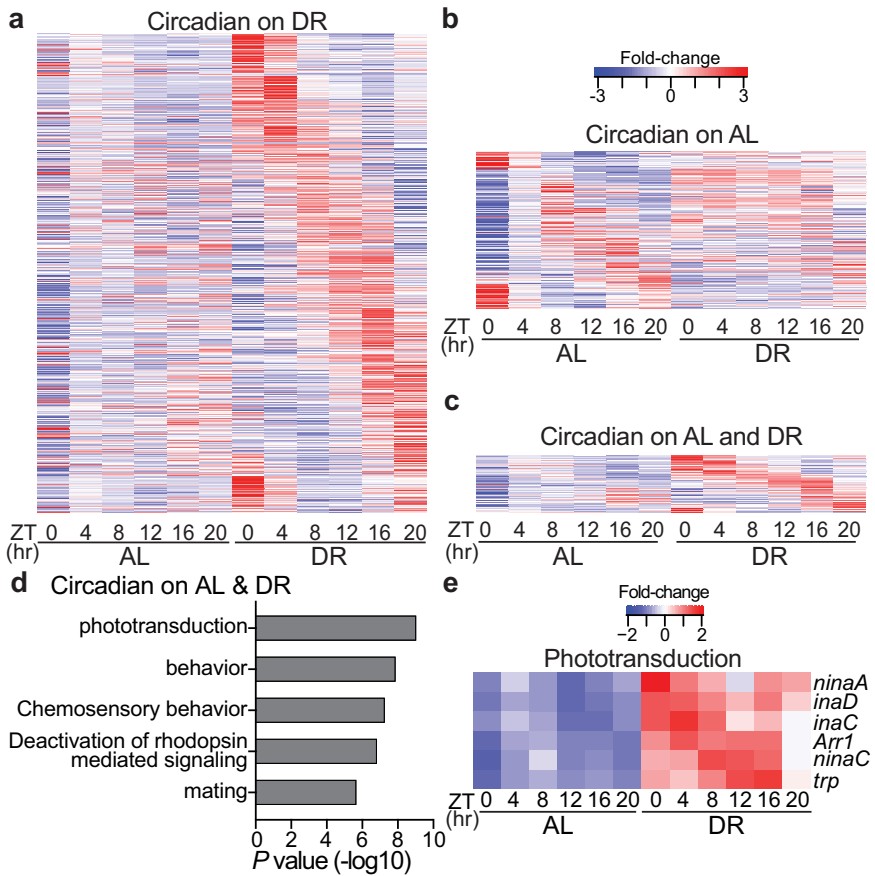

**Fig. 1 Dietary restriction amplifies circadian transcriptional output and rhythmicity of phototransduction genes. a–c** Circadian transcriptome heatmaps for *Canton-S* flies representing 24 h expression plots for transcripts that cycle only on DR (**a**, *n* = 1487 transcripts), only on AL (**b**, *n* = 548 transcripts), or on both diets (**c**, *n* = 259 transcripts). Circadian transcripts plotted here are defined as having JTK_CYCLE *P* ≤ 0.05 (non-adjusted, non-rhythmic [*P* ≥ 0.05] in *tim01* mutants) and are plotted by phase. **d** Gene-ontology enrichment categories corresponding to transcripts that cycle on both AL and DR diets. *P*-values were calculated with hypergeometric distribution (findGO.pl, HOMER) with no adjustment for multiple-hypothesis testing. **e** Heatmap of phototransduction transcript expression on AL and DR.

In *Drosophila* and mammals, visual function oscillates to align with daily changes in ambient illuminance from the sun, which can be $10^6$–$10^8$-fold brighter during the day than at night[17]. Photoreceptors are unique in that they have evolved mechanisms responsible for maintaining homeostasis in the presence of light-induced calcium ion gradients that are magnitudes greater than what other neuronal populations experience[18,19]. Mechanisms of light adaptation within photoreceptors include the rapid (millisecond) closure of TRP channels (facilitated via enzymes scaffolded by INAD), rhodopsin internalization from the rhabdomere membrane (e.g., ARR1, ARR2), and calcium efflux (e.g., CALX)[20,21]. Acrophase analyses (time of peak expression) revealed that circadian transcripts whose gene products encode proteins that promote photoreceptor activation ($Ca^{2+}$ influx) reach peak expression during the dark phase, while genes that encode proteins that terminate the phototransduction response (i.e., deactivation of rhodopsin-mediated signaling) peak in anticipation of the light phase (Supplementary Fig. 1l). These findings provide a potential mechanistic explanation for the rhythmic response pattern in light sensitivity observed in *Drosophila* photoreceptors[22] (See Supplementary Discussion 1 for additional interpretations).

To determine if molecular clocks mediate the enhanced expression of phototransduction genes on DR, we measured the transcriptome of fly heads with pan-neuronal overexpression of a dominant-negative form of the core-clock factor, CLK (Elav-GS-

GAL4 > UAS-CLK-Δ1; denoted nCLK-Δ1) (Supplementary Fig. 2a). To avoid potential developmental defects related to altered CLK function, we used the drug-inducible (RU486) GeneSwitch driver to express CLK-Δ in adult flies. Genes downregulated in nCLK-Δ1 heads were enriched for light-response pathways, including "response to light stimulus" and "deactivation of rhodopsin signaling" (Fig. 2a, b and Supplementary Data 3). Additionally, genes that were both circadian in wild-type heads and downregulated in nCLK-Δ1 were highly enriched for homeostatic processes related to eye function, while down-regulated genes in nCLK-Δ1 that were non-circadian in wild-type heads displayed no such enrichment (Supplementary Fig. 2c and Supplementary Data 4). Together, this indicates that CLK governs the transcriptional regulation of many eye-related processes in *Drosophila*.

Given DR's ability to improve homeostasis across an array of tissues[23], and its ability to enhance the circadian rhythmicity of light-response genes, we examined how diet and CLK influence visual function with age. We longitudinally quantified the positive phototaxis response of wild-type flies (*Canton-S* and *Oregon-R*) reared on either AL or DR diets (experimental setup in Supplementary Fig. 3a). A full description of the positive phototaxis responses and accompanied statistics for all strains used in this study are in Supplementary Data 5. Compared to AL-fed flies, DR slowed the decline in positive phototaxis observed with age (Supplementary Fig. 3b, c). Importantly, this effect

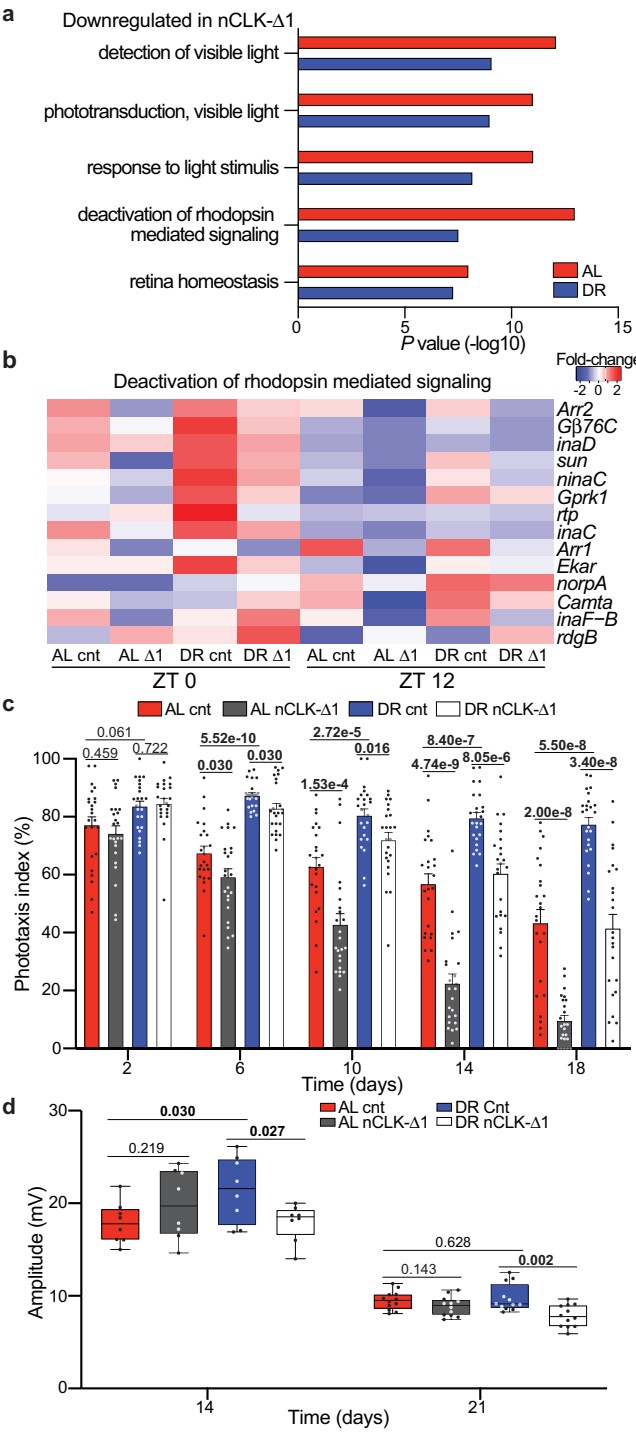

**Fig. 2 Dietary restriction delays visual senescence in a CLK-dependent manner. a** GO enrichment scores corresponding to downregulated light-response genes in heads from RNA-Seq of nCLK-Δ1 (Elav-GS-GAL4 > UAS-CLK-Δ1) vs controls. *P*-values were calculated with hypergeometric distribution (findGO.pl, HOMER) with no adjustment for multiple-hypothesis testing. **b** Heatmap of normalized RNA-Seq expression corresponding to the gene-ontology category "Deactivation of rhodopsin-mediated signaling" (GO:0016059) in nCLK-Δ1 and controls at zeitgeber times 0 and 12 (lights on and lights off, respectively). **c** Positive phototaxis responses for nCLK-Δ1 flies. For each timepoint results are represented as mean values of percent positive phototaxis ±SEM (*n* = 24 biologically independent cohorts of 20 flies examined over three independent experiments, *N* = 480 flies per condition). *P*-values were determined by two-tailed Student's *t*-test (unpaired), comparing responses between diet and/or genotype at each timepoint. **d** Boxplots of electroretinogram amplitudes for nCLK-Δ1 flies and controls at day 14 and 21. Data are presented as Tukey multiple comparison of means: The horizontal line within each box is the median, the bottom and top of the box are lower and upper quartiles, and the whiskers are minimum and maximum values. (*n* = 8 and 12 biologically independent flies measured at day 14 and 21, respectively, examined over 1 independent experiment). *P*-values were determined by two-tailed Student's *t*-test (unpaired), comparing responses between diet and/or genotype at each timepoint. Source data are provided as a Source Data file.

(an additional dominant-negative CLK mutant, Elav-GS-GAL4 > UAS-CLK-Δ2) flies displayed accelerated declines in positive phototaxis with age compared to controls (Fig. 2c and Supplementary Fig. 3f).

Since the positive phototaxis assay measures a behavioral response to light, we next evaluated how diet and CLK directly influence photoreceptor function with age by performing extracellular electrophysiological recordings of the eye (electro-retinograms, ERG[25]). A full description of the ERG responses and accompanied statistics for the fly strains used in this study are in Supplementary Data 6. We observed larger ERG amplitudes, i.e., the light-induced summation of receptor potentials from the photoreceptors[26], in control flies reared on DR vs AL on day 14 (Fig. 2d). Furthermore, the DR-mediated enhancements in the ERG amplitudes were significantly reduced in nCLK-Δ1 flies with age (Fig. 2d).

Since the Elav-GS-GAL4 driver is expressed in a pan-neuronal fashion (i.e., photoreceptors and extra-ocular neurons), we sought to examine how manipulating CLK function solely within photoreceptors influences visual function with age. To this end, we crossed UAS-CLK-Δ1 mutant flies and flies that over-express wild-type CLK (UAS-*clk*, denoted CLK-OE) with a photoreceptor-specific GAL4 driver line under the temporal control of the temperature-sensitive GAL80 protein (Trpl-GAL4; GAL80^{ts} > UAS-CLK-Δ1, denoted prCLK-Δ1 and Trpl-GAL4; GAL80^{ts} > UAS-CLK, denoted prCLK-OE). To avoid potential alteration of normal CLK function during development, prCLK-Δ1 and prCLK-OE flies were raised at 18 °C (GAL80 active, GAL4 repressed) and then transferred to 30 °C (GAL80 repressed, GAL4 active) following eclosion. When compared to control flies (Trpl-GAL4/+; GAL80^{ts}/+), prCLK-Δ1 flies displayed accelerated declines in both positive phototaxis and ERG amplitude with age, while prCLK-OE flies displayed a slower rate of visual senescence (Fig. 3a, b and Supplementary Fig. 4b). By day 14, prCLK-OE flies displayed a clear DR-mediated improvement in positive phototaxis compared to prClk-OE flies reared on AL (Fig. 3a). Interestingly, we failed to observe significant differences in diet-dependent responses (AL vs. DR) in the ERG's of prCLK-OE flies at any timepoint (Fig. 3b),

cannot solely be attributed to diet-dependent changes in locomotor activity, as climbing activity and phototaxis declined at different rates with age (Supplementary Fig. 3d). Compared to wild-type flies, DR minimally protected *clk^{out}* (*clk*-null) flies from age-related declines in phototaxis (Supplementary Fig. 3e). Unlike the *clk^{out}* mutant line, DR enhanced positive phototaxis responses in flies with mutations in the gene encoding the blue-light-sensitive Cryptochrome protein (*cry^{01}*, *cry^{02}*, and *cry^{B}* mutants) (Supplementary Fig. 3g–i). This discrepancy between diet-dependent responses to phototaxis in core-molecular clock mutants shows that DR's ability to improve phototaxis responses is not reliant on CRY's ability to photo-entrain the molecular clock[24], while CLK function is required. nCLK-Δ1 and nCLK-Δ2

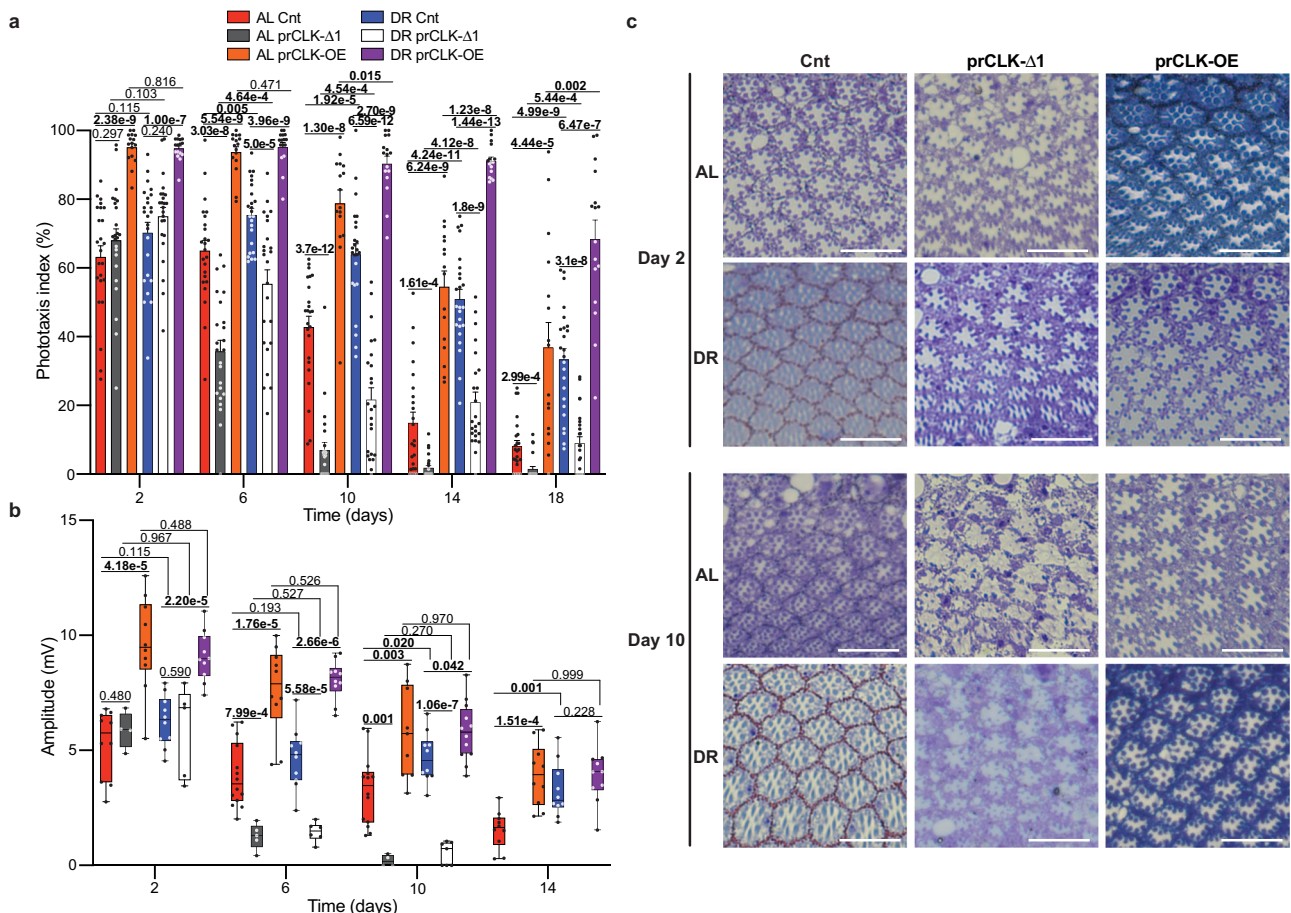

**Fig. 3 Photoreceptor clocks modulate visual function and degeneration with age. a** Positive phototaxis responses for prCLK-Δ1 flies (Trpl-GAL4; GAL80ts > UAS-CLK-Δ1), prCLK-OE (Trpl-GAL4; GAL80ts > UAS-*Clk*), and control flies (Trpl-GAL4; GAL80ts /+ ) reared at 30 °C. For each timepoint results are represented as mean values of percent positive phototaxis ±SEM (control and prCLK-Δ1: $n = 24$ biologically independent cohorts of 20 flies examined over three independent experiments, $N = 480$ flies per condition. prCLK-OE: $n = 16$ biologically independent cohorts of 20 flies examined over 2 independent experiments, $N = 320$ flies per condition). *P*-values were determined by two-tailed Student's *t*-test (unpaired), comparing responses between diet and/or genotype at each timepoint. **b** Boxplots of electroretinogram amplitudes for prCLK-Δ1, prCLK-OE, and control flies reared at 30 °C. The sample sizes (*n*) corresponding to (**b**) can be found in the "Statistics and Reproducibility" section within the Methods. Data are presented as Tukey multiple comparison of means: The horizontal line within each box is the median, the bottom and top of the box are lower and upper quartiles, and the whiskers are minimum and maximum values. *P*-values were determined by two-tailed Student's *t*-test (unpaired), comparing responses between diet and/or genotype at each timepoint. **c** Tangential sections through prCLK-Δ1, prCLK-OE, and control retinas at day 2 and day 10. R1-7 photoreceptors are apparent within each hexongonally shaped ommatidia. White bars in bottom right corners represents 10 microns. The number of biologically independent replicates for each group within this experiment are in the Statistics and Reproducibility section. Additional representative images are included with the source data file. Source data are provided as a Source Data file.

suggesting that the reduced phototaxis responses on AL diets observed in these lines were likely caused by behavioral and/or extra-ocular functional changes with age. We next visualized tangential sections of the eye to determine if the differences in phototaxis and ERG we observed via diet and genotype (prCLK-Δ1 and prCLK-OE) were associated with morphological changes to the ommatidium and photoreceptors. We failed to observe major morphological changes between genotypes on day 2, indicating that these groups developed a normal ommatidial structure, with the rhabdomere of the R1-7 photoreceptors clearly visible with no apparent degeneration (Fig. 3c). In agreement with the positive phototaxis and ERG data, we observed massive photoreceptor degeneration in the prCLK-Δ1 flies, while the control and prCLK-OE flies displayed normal ommatidia structure and rhabdomeres (Fig. 3c). Together, the gene expression, phototaxis, ERG, and morphological observations indicate that DR functions in a CLK-dependent manner to delay photoreceptor aging in the fly.

**nCLK-Δ drives a systemic immune response and reduces longevity.** Age-related declines in tissue homeostasis are accompanied by elevated immune responses and inflammation[27,28]. Interestingly, we found that genes upregulated in nCLK-Δ1 fly heads were significantly enriched for immune and antimicrobial humoral responses (Fig. 4a, b). In *Drosophila*, damage-associated molecular patterns can induce a sterile immune response that is characterized by the expression of antimicrobial peptides (AMPs), similar to the effects of infections by pathogens[29]. We quantified the mRNA expression of AMPs in the bodies of nCLK-Δ1 and nCLK-Δ2 flies to determine if neuronal damage signals propagate throughout the body to drive systemic immune responses; the *Drosophila* fat body generates high levels of AMPs in response to intrinsic damage signals[29]. The RT-PCR primer sequences used in this study are in Supplementary Data 7. AMP expression (*attA*, *diptB*, and *dro*) was reduced in control flies reared on DR compared to AL; however, expression of nCLK-Δ1 and nCLK-Δ2 elevated AMP expression on DR (Fig. 4c and

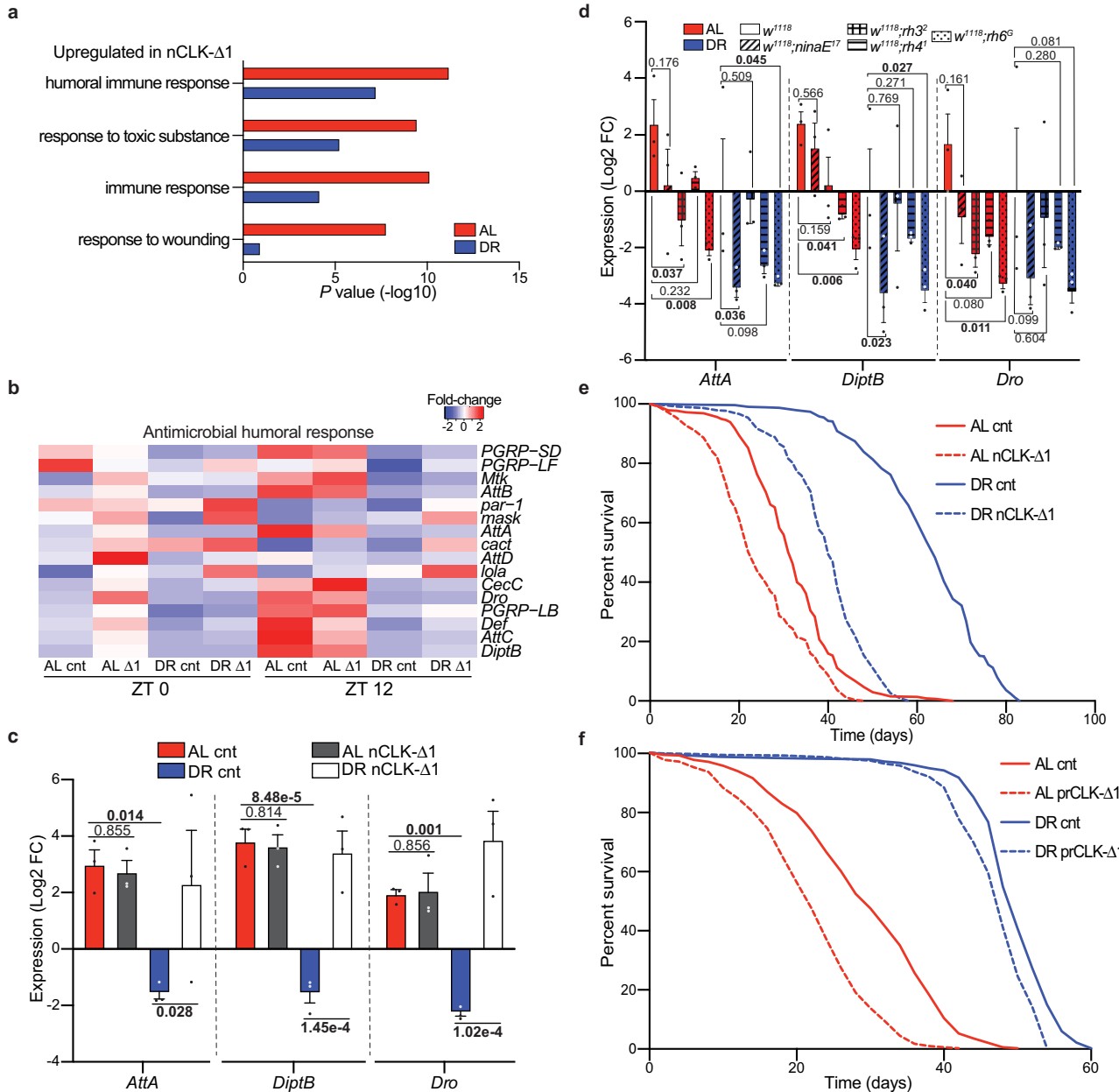

**Fig. 4 nCLK-Δ1 flies display elevated immune responses and shortened lifespan. a** GO enrichment scores corresponding to upregulated inflammatory genes in heads from RNA-Seq of nCLK-Δ1 vs controls. *P*-values were calculated with hypergeometric distribution (findGO.pl, HOMER) with no adjustment for multiple-hypothesis testing. **b** Heatmap of normalized RNA-seq expression corresponding to the gene-ontology category "Antimicrobial humoral responses" (GO:0019730) in nCLK-Δ1 and controls. **c** Relative expression of AMP genes (*attA*, *diptB*, and *dro*) calculated by RT-qPCR with mRNA isolated from nCLK-Δ1 bodies. Results are plotted as average Log2 fold-change in expression calculated by the ΔΔ-Ct method, normalized to DR vehicle-treated control samples, as well as the housekeeping gene *rp49* ± SEM (*n* = 3 biologically independent cohorts of flies, *N* = 30 flies per cohort). *P*-values were determined by two-tailed Student's *t*-test (unpaired), comparing Log2 fold-changes in expression. **d** Relative mRNA expression of immune genes (*attaA*, *diptB*, and *dro*) calculated by RT-qPCR with mRNA isolated from bodies of *w1118* and rhodopsin mutant flies housed in 12:12 h LD. Results are plotted as average Log2 fold-change in expression calculated by the ΔΔ-Ct method normalized *w1118* DR control samples as well as *rp49* ± SEM (*n* = 3 biologically independent cohorts of flies, *N* = 30 flies per cohort). *P*-values were determined by two-tailed Student's *t*-test (unpaired), comparing Log2 fold-changes in expression. **e** Kaplan–Meyer survival analysis of nCLK-Δ1 flies (Elav-GS-GAL4 > UAS-CLK-Δ1) reared at 25 °C. Survival data are plotted as an average of three independent lifespan repeats. Control flies (vehicle-treated): AL *N* = 575, DR *N* = 526; nCLK-Δ1 flies (RU486 treated): AL *N* = 570, DR *N* = 565. **f** Kaplan–Meyer survival analysis of prCLK-Δ1 flies (Trpl-GAL4; GAL80ts > UAS-CLK-Δ1) and control (Trpl-GAL4/+; GAL80ts/+), flies reared at 30 °C. Survival data are plotted as an average of three independent lifespan repeats. Control flies: AL *N* = 599, DR *N* = 501; prCLK-Δ1 flies: AL *N* = 513, DR *N* = 564. Source data are provided as a Source Data file.

Supplementary Fig. 5a). To further investigate this systemic inflammatory response, we isolated and quantified hemolymph from nCLK-Δ1 and control flies. In agreement with the transcriptional activation of AMPs in both the heads and bodies of nCLK-Δ1 flies, we found the most highly upregulated protein in nCLK-Δ1 hemolymph to be the antimicrobial peptide, AttC (Supplementary Fig. 5b). Furthermore, we observed an enrichment for proteins associated with translational activation (e.g., cytoplasmic translation and ribosomal biogenesis) within the upregulated proteins in the nCLK-Δ1 hemolymph, which may reflect the activation of hemocytes, the immune effector cells in *Drosophila* (Supplementary Data 8)[30]. Taken together, these data demonstrate that disrupting neuronal CLK function elevates systemic immune responses. However, as with all tissue-specific driver systems, we cannot rule out the possibility that our ELAV-GS-GAL4 driver expresses in a small population of non-neuronal cell types, which, in theory, could contribute to the elevated systemic inflammatory responses and/or influence lifespan.

To determine if photoreceptor degeneration induces a systemic immune response in *Drosophila*, we forced photoreceptor degeneration by knocking down *ATPα* within the eye (GMR-GAL4 > UAS-*ATPα*-RNAi), and quantified the expression of AMPs within the bodies. The *ATPα* gene encodes the catalytic alpha subunit of the $Na^+K^+$ATPase responsible for reestablishing ion balance in the eye during light responses[31,32]. Our decision to use *ATPα* knockdown as a model of photoreceptor degeneration was motivated by previous reports indicating that its expression is under circadian regulation[33] and that its knockdown in the eye results in aberrant ion homeostasis that drives age-dependent, light-independent photoreceptor degeneration[34]. Ocular knockdown of *ATPα* rendered flies blind in both AL and DR conditions compared to controls (Supplementary Fig. 5c). Knocking down *ATPα* in the eye also drove the expression of AMPs within the bodies of flies reared on either an AL or DR diet (Supplementary Fig. 5d). Thus, DR fails to suppress immune responses in the context of forced photoreceptor degeneration.

Since we found that photoreceptor degeneration induced systemic immune responses, we postulated that reducing phototransduction should reduce inflammation. To assess how stress from environmental lighting influences immune responses, we analyzed a circadian microarray dataset comparing gene expression changes in wild-type (y,w) heads in flies reared in 12 h light and 12 h darkness (12:12 LD) or constant darkness[13]. We found immune response genes to be among the most highly enriched processes upregulated in the flies housed in 12:12 LD vs constant darkness (Supplementary Fig. 5e, f and Supplementary Data 9). We quantified AMPs within the bodies of flies harboring rhodopsin-null mutations to evaluate how the different photoreceptor subtypes influence systemic immune responses. The *Drosophila* ommatidia consist of eight photoreceptors (R1-8) that express different rhodopsins with varying sensitives to distinct wavelengths of light[35]. The R1-6 photoreceptors express the major rhodopsin Rh1, encoded by *ninaE*, while the R7 photoreceptor expresses either Rh3 or Rh4. The R8 photoreceptor expresses either Rh5 or Rh6[36]. The rhodopsin-null mutants [*ninaE*[1737], *rh3*[238], *rh4*[139], or *rh6*[G40]] displayed reductions in immune marker expression in their bodies compared to *w*[1118] outcrossed controls (Fig. 4d). Taken together, these findings indicate that suppression of rhodopsin-mediated signaling is sufficient to suppress systemic immune responses in *Drosophila* housed in LD conditions.

Given the strong associations between chronic immune activation and accelerated aging, we measured the lifespans of nCLK-Δ flies[27]. Both nCLK-Δ1 and nCLK-Δ2 flies displayed significantly shortened lifespans, with a proportionally greater loss in median lifespan in flies reared on DR compared to AL

(Fig. 4e and Supplementary Fig. 6a–c). Detailed information on the survival analyses performed in this study and accompanied statistics are in Supplementary Data 10. nCLK-Δ flies have altered CLK function throughout all neurons; however, it is possible that the lifespan-shortening effect observed in these lines was substantially driven by the loss of CLK function within the eye; Others have demonstrated that CLK is highly enriched (>5-fold) within photoreceptors compared to other neuronal cell types in *Drosophila* (Supplementary Fig. 6d)[5]. Furthermore, overexpressing CLK-Δ1 within just photoreceptors (prCLK-Δ1) also shortened lifespan (Fig. 4f). These findings argue that neuronal CLK function is required for the full lifespan extension mediated by DR and indicate that the function of the transcription factor CLK within photoreceptors is essential for the maintenance of visual function with age as well as organismal survival.

**DR protects against lifespan shortening from photoreceptor cell stress.** Previous reports have demonstrated that exposure to light can decrease lifespan—extending the daily photoperiod, or housing flies in blue light both reduce longevity[41,42]. Since DR delays visual senescence and promotes the rhythmic expression of genes involved in photoreceptor homeostasis (i.e., light adaptation, calcium handling), we investigated how diet influences survival in the context of light and/or phototransduction. To test the interrelationship between diet, light, and survival, we housed *w*[1118] (white-eyed) in either a 12:12 LD cycle or constant darkness. Housing flies in constant darkness extended the lifespan of flies reared on AL, while the lifespans of flies reared on DR were unaffected (Fig. 5a). Constant darkness failed to extend the lifespan of red-eyed (*w*[+]) Canton-S wild-type flies (Supplementary Fig. 7a), suggesting that the ATP-binding cassette transporter encoded by *w*, and the red-pigment within the cone cells, help to protect against lifespan shortening from diet- and light-mediated stress[43]. White-eyed, photoreceptor null flies (homozygous for *TRP*[P365] mutation[44]) reared on AL failed to display lifespan extension in constant darkness (Supplementary Fig. 7b), indicating that the lifespan-shortening effects of light exposure are primarily mediated by the photoreceptors.

We performed survival analyses in rhodopsin-null flies to examine how activation of the different photoreceptor subtypes influences lifespan on AL and DR. In agreement with the reduction in systemic immune responses observed in the rhodopsin-null strains, these flies were also longer lived in comparison to *w*[1118] outcrossed controls (Fig. 5b–e). Furthermore, *rh6*[G] mutants, which displayed the largest reductions in inflammation, also displayed the greatest extension in lifespan compared to the other rhodopsin-null lines. *Gqα*[1] mutant flies[45], which harbor a mutation in the G-protein that mediates activation of TRP channels downstream of rhodopsin, also displayed increased longevity compared to control flies (Fig. 5f). Interestingly, with the exception of *rh4*[1], rhodopsin-null mutations and *Gqα*[1] mutants primarily extended lifespan on AL, indicated by the hazard ratios in Fig. 5g. We next sought to investigate how increases in rhodopsin-mediated signaling influence survival. To this end, we knocked down the major arrestin, encoded by *arr1*, within the eyes of flies (GMR-GAL4 > UAS-*arr1*-RNAi). ARR1 is required for light-mediated rhodopsin internalization from the rhabdomere membrane into endocytic vesicles, thus suppressing rhodopsin-mediated signaling and associated $Ca^{2+}$-mediated phototoxicity/cell death[20,46,47]. *Arr1* mRNA expression is circadian in wild-type heads (Supplementary Fig. 1m) and is also a direct CLK target (CLK ChIP-Seq., Supplementary Table 1). Additionally, *arr1* expression was significantly downregulated in heads of nCLK-DN1 flies compared to controls (Supplementary Fig. 7c). In agreement with its

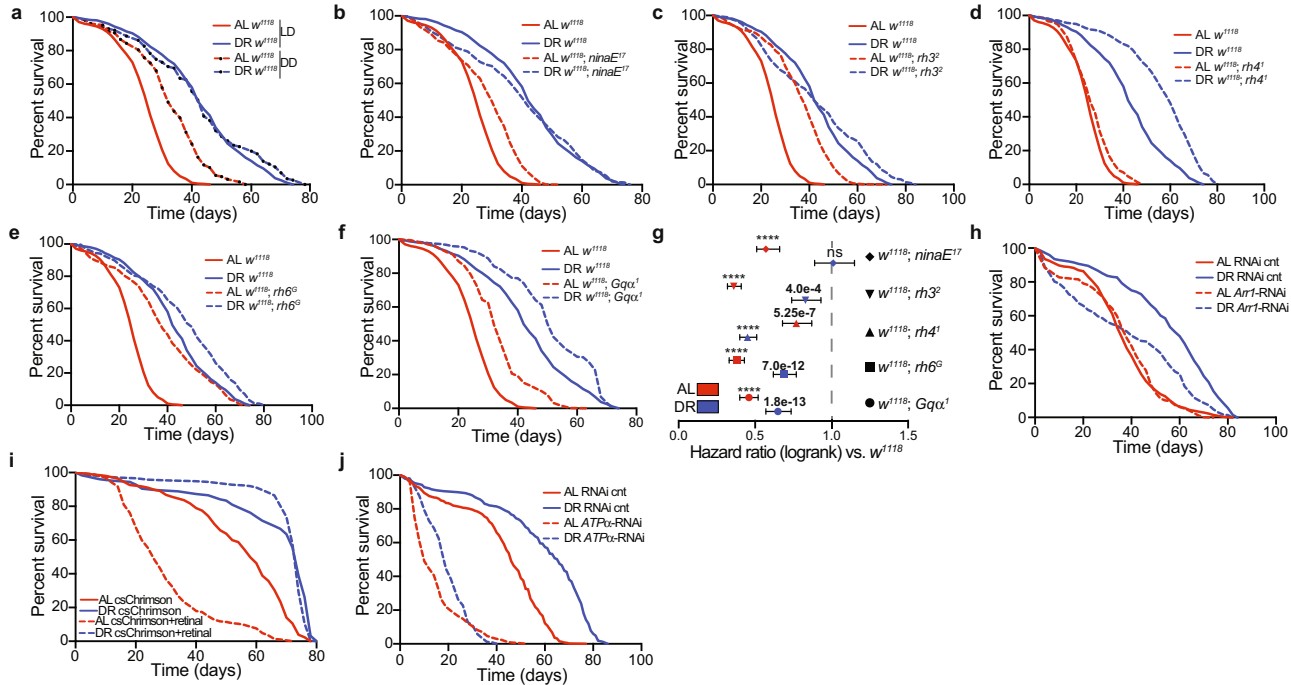

**Fig. 5 Photoreceptor activation modulates lifespan in a diet-dependent fashion. a** Survival analysis of $w^{1118}$ flies housed in 12:12 h LD or constant darkness (DD). Survival data are plotted as an average of three independent lifespan repeats. **b–f** Survival analysis of $w^{1118}$; $ninaE^{17}$, $w^{1118}$; $rh3^2$, $w^{1118}$; $rh4^1$, $w^{1118}$; $rh6^G$, and $w^{1118}$; $Gq\alpha^1$ mutants compared to $w^{1118}$ control flies housed in 12:12 h LD. Survival data are plotted as an average of three independent lifespan repeats. *Survival curves for $w^{1118}$ are re-plotted (**b–f**) for visual comparison, and the $w^{1118}$ and rhodopsin-null lifespans repeats were performed simultaneously. All mutant lines were outcrossed to $w^{1118}$. **g** Hazard ratios for rhodopsin and Gq mutant flies compared to $w^{1118}$ control flies (ratios < 1 indicate flies that are more likely to survive compared to $w^{1118}$). The hazard ratio for each strain is plotted as the measure of centre and the error bars indicate the 95% confidence interval of the hazard ratios. $P$-values were determined by Log-rank (Mantel-Cox) test, ns denotes a non-significant $P$-value, **** indicates $P \le 1.0e\text{-}15$. **h** Survival analysis of eye-specific $arr1$-RNAi knockdown flies vs RNAi control flies. Survival data are plotted as an average of two independent lifespan repeats for $arr1$-RNAi and one independent lifespan replicate for RNAi-controls. **i** Survival analysis of retinal inducible, photoreceptor-specific optogenetic flies (Trpl-GAL4 > UAS-csChrimson [red-shifted]) supplemented with retinal or vehicle control and housed in 12:12 h red light:dark. Survival data are plotted as an average of two independent lifespan repeats. **j** Survival analysis of eye-specific $ATP\alpha$ RNAi knockdown flies vs RNAi control flies. Survival data are plotted as an average of three independent lifespan repeats. The total number of flies ($N$) corresponding to each lifespan in Fig. 5 can be found in the "Statistics and Reproducibility" section within the Methods. Source data are provided as a Source Data file.

physiological role in light adaptation, we found that $arr1$-RNAi knockdown flies were hypersensitized to light (Supplementary Fig. 7d). In contrast to the rhodopsin-null strains, which displayed greater proportional improvements in survival on AL vs DR, $arr1$-RNAi knockdown flies displayed a significant shortening of lifespan on DR, while the lifespan on AL was indistinguishable from the control (Fig. 5h). Together, these findings suggest that modulation of rhodopsin-mediated signaling is sufficient to regulate lifespan in *Drosophila*.

Next, we used an optogenetics approach to examine how chronic photoreceptor activation influences survival in flies reared on AL or DR. This approach allowed us to directly compare the survival of isogenic flies with or without photoreceptor activation, while avoiding the potentially confounding effect that extra-ocular light-sensing might have on survival. Namely, we expressed the red-light-sensitive csChrimson cation channel[48] within photoreceptors (Trpl-GAL4 > UAS-csChrimson). To activate the csChrimson channels, we housed the optogenetic flies in a 12:12 red light:dark cycle and supplemented their food with either all-*trans* retinal (a chromophore required for full activation of csChrimson channels[49]) or vehicle control (Supplementary Fig. 7e). Optogenetic activation of the photoreceptors (retinal-treated) drastically reduced AL lifespan compared to vehicle-treated controls, while the lifespan on DR was unaffected (Fig. 5i). Retinal did not appear to be toxic to flies lacking csChrimson channels, as the lifespan of *Canton-S* wild-

type flies was indistinguishable between vehicle- and retinal-treated groups (Supplementary Fig. 7f).

Although DR protected flies from lifespan shortening from the optogenetic activation of photoreceptors, we found that forcing photoreceptor degeneration, by knocking down $ATP\alpha$ in the eye (GMR-GAL4 driver) shortened the lifespan of both AL and DR (Fig. 5j). In addition to knocking down $ATP\alpha$ throughout all eye cell types, we also targeted $ATP\alpha$ expression specifically within the cone cells that surround the photoreceptors (Spa-GAL4 > UAS-$ATP\alpha$-RNAi). We found that knockdown of $ATP\alpha$ mRNA within cone cells was also sufficient to accelerate visual declines with age and reduce the lifespan in flies reared on AL and DR (Supplementary Fig. 8a, b). Similarly, eye-specific knockdown (achieved via GMR-Gal4) of Nervana 2 ($nrv2$-RNAi) and Nervana 3 ($nrv3$-RNAi), which encode the *Beta* subunit of the $Na^+K^+$ATPase of the eye[34] also reduced phototaxis responses and shortened lifespan (Supplementary Fig. 8c–f). Taken together, these data support a model where DR protects flies from lifespan shortening caused by photoreceptor stress, as chronic photoreceptor activation reduces survival in flies reared on AL while having minimal to no effect on flies reared on DR. Inversely, photoreceptor deactivation primarily improves survival of flies reared on AL.

**Eye-specific, CLK-output genes modulate lifespan.** We next sought to determine if CLK-output genes in the eye influence age-

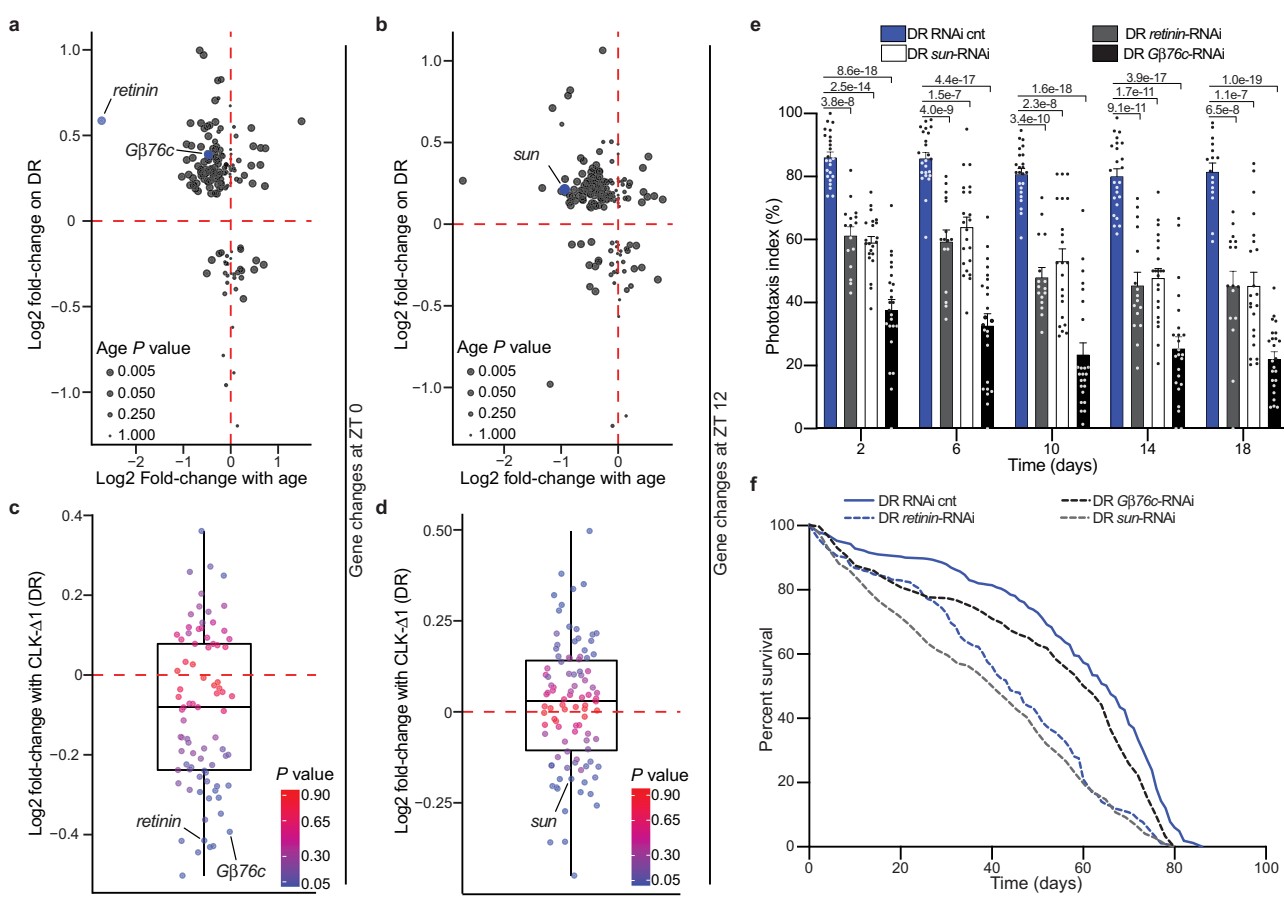

**Fig. 6 Knockdown of DR-sensitive, eye-specific CLK-output genes reduces survival. a, b** Scatterplot of circadian, photoreceptor-enriched gene changes with age in wild-type heads (x-axis: 5- vs 55-day old flies) vs diet-dependent gene expression changes in heads from nCLK-Δ1 RNA-Seq control flies (y-axis: DR- vs AL-minus RU486) at ZT 0 (**a**) and ZT 12 (**b**). Age P-values (non-adjusted) were originally reported in ref. [16] and were calculated with Cuffdiff comparing transcript expression at day 5 to day 55 in wild-type *Canton-S* heads. **c, d** Boxplots of the expression changes in nCLK-Δ1 heads (DR plus- vs DR minus-RU486) at ZT 0 (**c**) and ZT 12 (**d**) of genes that were downregulated with age and upregulated on DR (upper left quadrants of panels **a, b**). The horizontal line within each box is the median, the bottom and top of the box are lower and upper quartiles, and the whiskers are minimum and maximum values. P-values (non-adjusted) were calculated with DESeq2 comparing transcript expression between controls and nCLK-Δ1 flies reared on DR. **e** Positive phototaxis responses with eye-specific knockdown of Gβ76c (GMR-GAL4 > UAS-Gβ76c-RNAi), retinin (GMR-GAL4 > UAS-retinin-RNAi), and sun (GMR-GAL4 > UAS-sun-RNAi) compared to RNAi control flies (GMR-GAL4 > UAS-mCherry-RNAi) reared on DR. For each timepoint results are represented as mean values of phototaxis responses ±SEM (RNAi control n = 24 biologically independent cohorts of 20 flies examined over three independent experiments, N = 480 flies per condition; Gβ76c-RNAi n = 24 biologically independent cohorts of 20 flies examined over three independent experiments, N = 480 flies per condition; retinin RNAi n = 16 biologically independent cohorts of 20 flies examined over two independent experiments, N = 320 flies per condition; sun RNAi n = 24 biologically independent cohorts of 20 flies examined over three independent experiments, N = 480 flies per condition) P-values were determined by two-tailed Student's t-test (unpaired) at each timepoint comparing the phototaxis index of RNAi control flies to Gβ76c-, retinin-, and sun-RNAi flies. **f** Survival analysis of eye-specific Gβ76c, retinin, sun, and RNAi knockdown flies compared to RNAi control flies reared on DR. Survival data are plotted as an average of three independent lifespan repeats for RNAi control, sun, and Gβ76c flies and two independent lifespan repeats for retinin RNAi knockdown flies. RNAi-cnt flies: N = 490; retinin-RNAi flies: N = 363; sun-RNAi flies: N = 468; Gβ76c-RNAi flies: N = 509. Source data are provided as a Source Data file.

related visual declines and lifespan. We employed a bioinformatics approach to identify candidate eye-specific circadian genes transcriptionally regulated by CLK (Supplementary Fig. 9a and Supplementary Data 11). First, we compared age-associated changes in photoreceptor-enriched gene expression[5] to genes that were differentially expressed on DR compared to AL. More than half of the photoreceptor-enriched genes that were downregulated with age were also upregulated on DR at ZT 0 and ZT 12 (upper left quadrant of Fig. 6a, b and Supplementary Fig. 9a). We then subset this gene list, selecting just transcripts whose expression was downregulated with age and upregulated on DR, and examined how their expression changed in nCLK-Δ1 fly heads (Supplementary Fig. 9a). From this analysis, we identified G-protein beta-subunit 76 C (Gβ76c), retinin, and tetraspanin

42Ej (sun) as genes that were significantly downregulated in nCLK-Δ1 fly heads at ZT 0 and/or ZT 12 (Fig. 6c, d and Supplementary Fig. 9a, e–g). Gβ76c encodes the eye-specific G beta-subunit that plays an essential role in terminating phototransduction[14,50]. Retinin encodes one of the four most highly expressed proteins in the lens of the *Drosophila* compound eye[51]. Furthermore, Retinin functions in the formation of corneal nanocoatings, knockdown of which results in degraded nanostructures and a reduction in their anti-reflective properties[52]. Sun encodes for a lysosomal tetraspanin concentrated in the retina that protects against photoreceptor degeneration by degrading rhodopsin in response to light[53]. We analyzed a published CLK Chromatin Immunoprecipitation (ChIP-chip) dataset in flies and observed rhythmic CLK binding at the 5'-untranslated region of

*sun* in *Drosophila* eye tissue[54] (Supplementary Fig. 9h and Supplementary Table 1), which supports our bioinformatics approach and provides further evidence that *sun* is an eye-specific CLK-output gene. Eye-specific knockdown of *Gβ76c* (GMR-GAL4 > UAS-*Gβ76c*-RNAi)*, retinin* (GMR-GAL4 > UAS-*retinin*-RNAi), and *sun* (GMR-GAL4 > UAS-*sun*-RNAi) reduced phototaxis responses (Fig. 6e and Supplementary Fig. 9i), and shortened lifespan in comparison to RNAi control flies (GMR-GAL4 > UAS-mCherry-RNAi) (Fig. 6f, Supplementary Fig. 9j). These findings indicate that DR and CLK function together in the regulation of eye-specific circadian genes involved in the negative regulation of rhodopsin signaling (i.e., phototransduction termination). Furthermore, these observations support previous findings that lifespan extension upon DR requires functional circadian clocks[10,55], and establishes circadian CLK-output genes as diet-dependent regulators of eye aging and lifespan in *Drosophila*.

## Discussion

Progressive declines in circadian rhythms are one of the most common hallmarks of aging observed across most lifeforms[56]. Quantifying the strength, or amplitude, of circadian rhythms is an accurate metric for predicting chronological age[57]. Many cellular processes involved in aging (e.g., metabolism, cellular proliferation, DNA repair mechanisms, etc.) display robust cyclic activities. Both genetic and environmental disruptions to circadian rhythms are associated with accelerated aging and reduced longevity[58,59]. These observations suggest that circadian rhythms may not merely be a biomarker of aging; rather, declines in circadian rhythms might play a causal role. The observation that DR and DR-memetics, such as calorie restriction and time-restricted feeding, improve biological rhythms suggests that clocks may play a fundamental role in mediating their lifespan-extending benefits.

Herein, we identified circadian processes that are selectively amplified by DR. Our findings demonstrate that DR amplifies circadian homeostatic processes in the eye, some of which are required for DR to delay visual senescence and improve longevity in *Drosophila*. We report that disrupting CLK function within photoreceptors accelerates visual declines and shortens lifespan, while overexpressing wild-type CLK protects against age-associated declines in vision and rescues AL-dependent declines in photoreceptor function. Our data also demonstrate that photoreceptor stress has deleterious effects on organismal health; overstimulation of the photoreceptors induced a systemic immune response and reduced longevity.

Among the more interesting and unexpecting findings of this study is the observation that the *Drosophila* eye influences systemic immune responses, as we observed elevated AMP expression in the bodies of flies overexpressing CLK-Δ pan-neuronally and in flies with forced photoreceptor degeneration (*ATPα*-RNAi). It is possible that GAL4 misexpression may promote inflammatory responses in the fly bodies, although we found a reduction in systemic inflammation in the rhodopsin-null lines suggesting that this phenomenon can originate at the photoreceptor. We also found that these systemic immune responses correlate with lifespan changes (increased body AMP expression is associated with declines in longevity and vice versa), similar to what is observed with chronic inflammation or "inflammaging" in other models[60]. However, we cannot conclude whether neuronal or eye-mediated increases in systemic inflammation are causal to aging in other tissues. Furthermore, the mechanisms by which the *Drosophila* eye, and, more specifically, the photoreceptor influence systemic immune responses are unclear. We speculate that photoreceptor degeneration may disrupt the retinal-blood barrier such that damage signals from the eye

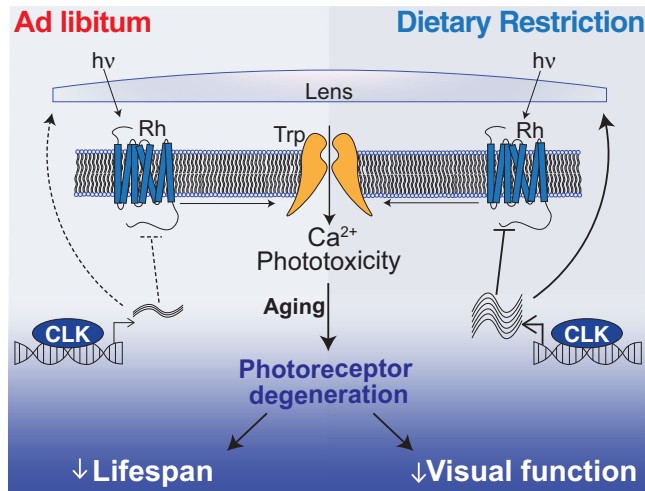

**Fig. 7 Dietary restriction extends lifespan by promoting rhythmic homeostatic processes in the eye.** DR promotes CLK-output processes in the eye that suppress light/$Ca^{2+}$-mediated phototoxicity to delay visual senescence and improve survival.

propagate through to the hemolymph and activate AMP expression in distal tissues. Future studies are aimed at elucidating this mechanism, and its effect on longevity.

Our findings establish the eye as a diet-sensitive regulator of lifespan. DR's neuroprotective role in the photoreceptors appears to be mediated via the transcription factor CLK, which promotes the rhythmic oscillation of genes involved in the suppression of phototoxic cell stress (Fig. 7 and Supplementary Discussion 1). Given that CLK transcriptionally regulates circadian and non-circadian transcripts, future investigations may determine whether the time-of-day regulation of these genes by CLK is germane to promoting eye health with age. These studies may also examine whether the DR-mediated benefits on visual senescence and photoreceptor viability are mediated solely by CLK as a transcription factor (as demonstrated here) or whether circadian clock function (rhythmic output) is required. Our findings also support the notion that age-related declines in the visual system impose a high cost on an organism's physiology. Perhaps this provides an alternative hypothesis for why several cave-dwelling animals, whose visual systems have undergone regressive evolution (e.g., cave-dwelling fish and naked-mole rats), are especially long-lived[61]. Failing to develop a visual system may act as a pro-survival mechanism allowing organisms to avoid the damage and inflammation triggered by age-related retinal degeneration. Ultimately, developing a visual system, which is critical for reproduction and survival, may be detrimental to an organism later in life. Thus, vision may be an example of an antagonistically pleiotropic mechanism that shapes lifespan.

## Methods

**Experimental materials**. A full detailed description of the materials, primer sequences, software packages, and commercial assays used in this study are reported in Supplementary Data 7. Data was collected with Microsoft Excel (version 16.58) and figures and statistics were generated in GraphPad Prism software (version 9).

**Fly stocks**. The genotypes of the *Drosophila Melanogaster* lines used in this study are listed in Supplementary Table 2. The following lines were obtained from the Bloomington *Drosophila* Stock Center: Oregon R. (25125), GMR-GAL4 (1104), Elav-GS-GAL4 (43642), Trpl-GAL4 (52274), *clk^{out}* (56754), UAS-csChrimson (55134), UAS-CLK-Δ1 (36318), UAS-CLK-Δ2 (36319), *Gβ76c*-RNAi (28507), *tsp42Ej/sun*-RNAi (29392), *retinin*-RNAi (57389), *ATPα*-RNAi (28073), *nrv2*-RNAi (28666), *nrv3*-RNAi (60367), and mCherry-RNAi (Bloomington RNAi-cnt, 35785). The following lines were obtained from the Vienna *Drosophila* Resource Center: *arr1*-RNAi (22196), *RNAi*-cnt (empty vector, 60100). The UAS-*Clk* line

was a gift from Paul Hardin's laboratory at Texas A&M University. The following lines were outcrossed to $w^{1118}$ for this manuscript: UAS-CLK-$\Delta1^{OC}$ and Canton-$S^{OC}$. The Trpl-GAL4 line was recombined with GAL80 for this manuscript: Trpl-GAL4; GAL80$^{ts}$. Ethical approval was not requested for the experiments performed in this study given the exclusive use of fruit flies.

**Age of fly strains used in this study.** Canton-S and $tim^{01}$ mutant flies (11 days old) were used for the time-course microarray experiment. nCLK-$\Delta1$ flies (11 days old) were used for the RNA-Seq. experiment. The following strains (6, 10, 14, 18, and 22 days old) were used in phototaxis experiments: nCLK-$\Delta1$, prCLK-cnt, prCLK-$\Delta1$, prCLK-OE, GMR-GAL4 > RNAi-cnt, GMR-GAL4 > Gbeta76c-RNAi, GMR-GAL4 > retinin-RNAi, GMR-GAL4 > sun-RNAi, Canton-S, Oregon-R, $clk^{out}$, nCLK-$\Delta2$, $cry01$, $cry02$, cryB, GMR-GAL4 > ATPα-RNAi, GMR-GAL4 > arr1-RNAi, GMR-GAL4 > nrv2-RNAi, GMR > GAL4 > nrv3-RNAi, Spa-GAL4 > RNAi-cnt, Spa-GAL4 > ATPα-RNAi. The follow strains (6, 10, 14, 18, and 25 days old) were used in ERG experiments: nCLK-$\Delta1$, prCLK-cnt, prCLK-$\Delta1$, prCLK-OE. The follow strains (6 and 14 days old) were used in tangential eye sections experiments: prCLK-cnt, prCLK-$\Delta1$, prCLK-OE. The following strains (11 days old) were used in RT-PCR experiments: nCLK-$\Delta1$, nCLK-$\Delta2$, $w^{1118}$, $ninaE^{17}$, $rh3^2$, $rh4^1$, $rh6^G$. The following strains (6-days old to death) were used for lifespan analyses: nCLK-$\Delta1$, nCLK-$\Delta2$, $w^{1118}$, $ninaE^{17}$, $rh3^2$, $rh4^1$, $rh6^G$, $Gq^1$, prCLK-cnt, prCLK-$\Delta1$, prCLK-OE, GMR-GAL4 > RNAi, GMR-GAL4 > Gbeta76c-RNAi, GMR-GAL4 > -retinin-RNAi, GMR-GAL4 > sun-RNAi, Canton-S, csChrimson, $TRP^{365}$, GMR-GAL4 > ATPα-RNAi, GMR-GAL4 > arr1-RNAi, GMR-GAL4 > nrv2-RNAi, GMR > GAL4 > nrv3-RNAi, Spa-GAL4 > RNAi-cnt, Spa-GAL4 > ATPα-RNAi. nCLK-$\Delta1$ flies at 18 days of age were used in the hemolymph mass-spectrometry experiment.

**Fly husbandry and survival analyses.** All flies were maintained at $25 \pm 1$ °C, 60% humidity under a 12 h:12 h LD cycle (~750lux, as measured with a Digital Lux Meter, Dr. Meter Model LX1330B) unless otherwise indicated. Fly stocks and crosses were maintained on a standard fly media which consisted of 1.5% yeast extract, 5% sucrose, 0.46% agar, 8.5% of cornmeal, and 1% acid mix (a 1:1 mix of 10% propionic acid and 83.6% phosphoric acid). Fly bottles were seeded with live yeast prior to collecting virgins or setting up crosses. Mated adult progeny were then transferred to ad libitum (AL) or dietary restriction (DR) media within three days of eclosion. Adult female flies used in experiments were transferred to fresh media every 48 h at which point deaths were recorded for survival analysis. AL and DR fly media differed only in their percentage of yeast extract, respectively containing 5% or 0.5% (Yeast Extract, B.D. Bacto, Thermo Scientific 212720, Cat no. 90000-722). Optogenetic experiments: For experiments using the csChrimson channel rhodopsin[48], adult flies were transferred to media supplemented with 50 μM all-trans-retinal (Sigma–Aldrich, R2500-1G) or drug vehicle (100% ethanol), and maintained under a 12 h:12 h red light:dark cycle, with ~10lux of red light (~590 nm) during the light phase. Elav-GeneSwitch flies: GeneSwitch[62], adult flies were transferred to media supplemented with 200 μM RU486 (Mifepristone, United States Biological), indicated as either AL+ or DR+, for post-developmental induction of transgenic elements; isogenic control flies were transferred to food supplemented with a corresponding concentration of drug vehicle (100% ethanol), indicated as either AL- or DR-. prCLK-$\Delta1$ experiments: GAL80 temperature-sensitive crosses were set in bottles at 25 °C, 60% humidity under a 12 h:12 h LD cycle for 4 days. Parental flies were removed, and the bottles were transferred to 18 °C for ~3 weeks to suppress GAL4 activity throughout development. After eclosion, the F1 generations were sorted onto AL or DR food the flies were maintained at 30 °C to de-repress GAL80 and activate GAL4 (60% humidity under a 12 h:12 h LD cycle) for the remainder of their lifespans. The F1 generations for these experiments share the same genetic background, as both the UAS-CLK-$\Delta1$ and the Canton-S control lines were fully outcrossed to the same $w^{1118}$ strain prior to setting up the cross with Trpl-GAL4; GAL80$^{ts}$.

**Circadian time-course expression analysis.** Mated Canton-S and $Tim^{01}$ females were reared on AL or DR diets for seven days at $25 \pm 1$ °C, under a 12:12 h light-dark (LD) regimen. Beginning on the seventh day, four independent biological replicates (per diet/timepoint) of approximately 35 female flies (approximately 11 days old) were collected on dry ice every 4 h for 20 h starting at ZT 0 (six total timepoints, 48 total samples). RNA extraction, DNA amplification/labeling, and gene expression arrays were performed following the same protocols as in ref. [63]. In summary, RNA was isolated from whole-fly lysates with Qiagen's RNeasey Lipid Tissue Mini Kit (74804), and RNA quantity and quality were accessed with a Nanodrop and Agilent's bioanalyzer (RNA 600 Nano Kit (5067-15811)). DNA amplification from total RNA was performed using Sigma's TransPlex Complete Whole Transcriptome Amplification Kit (WTA2) and purified with Qiagen's QIAquick PCR Purification Kit (28104). Gene expression labeling was performed with NimbleGen One-Color DNA Labeling Kit (05223555001) and hybridized to NimbleGen 12-Plex gene expression arrays. Arrays were quantitated with NimbleGen's NimbleScan2 software (version 2.6), and downstream expression analyses were conducted in R (version 3.2.4) (http://www.r-project.org) with the LIMMA package 3.34.5. Transcript-level expressions from the four independent biological replicates were averaged for each timepoint.

**nCLK-$\Delta1$ RNA-seq analyses.** nCLK-$\Delta1$ (Elav-GS-GAL4 > UAS-nCLK-$\Delta1$) adult female flies were developed on standard stock food (1.5% yeast-extract) for four days. Three independent biological replicates of 100 mated female flies were then reared on AL or DR diets treated with RU486 or vehicle control at $25 \pm 1$ °C, under a 12:12 h LD regimen. Diets were changed approximately every 48 h, until the seventh day at which point flies were flash-frozen (~11 days old) on dry ice at ZT 0 and ZT 12 (lights-on and -off, respectively). See Supplementary Fig. 2a for RNA-seq. experimental design. RNA extraction: Frozen flies were vortexed to remove heads and mRNA from each biological replicate of pooled heads was isolated with the Quick-RNA MiniPrep Kit (Zymo Research #11-328), per manufacturers' instructions. Fragment library preparation and deep sequencing: Library preparation was performed by the Functional Genomics Laboratory (FGL), a QB3-Berkeley Core Research Facility at University of California, Berkeley. cDNA libraries were produced from the low-input RNA using the Takara SMART-Seq v4 Ultra-low-input RNA kit. An S220 Focused-Ultrasonicator (Covaris®) was used to fragment the DNA, and library preparation was performed using the KAPA hyper prep kit for DNA (KK8504). Truncated universal stub adapters were used for ligation, and indexed primers were used during PCR amplification to complete the adapters and to enrich the libraries for adapter-ligated fragments. Samples were checked for quality on an AATI (now Agilent) Fragment Analyzer. Samples were then transferred to the Vincent J. Coates Genomics Sequencing Laboratory (GSL), another QB3-Berkeley Core Research Facility at UC Berkeley, where Illumina sequencing library molarity was measured with quantitative PCR with the Kapa Biosystems Illumina Quant qPCR Kits on a BioRad CFX Connect thermal cycler. Libraries were then pooled evenly by molarity and sequenced on an Illumina NovaSeq6000 150PE S4 flowcell, generating 25 M read pairs per sample. Raw sequencing data were converted into fastq format, sample-specific files using the Illumina bcl2fastq2 (v2.20) software on the sequencing center local linux server system. Read alignment and differential expression analyses: Raw fastq reads were filtered by the Trimmomatic software[64] (Trimmomatic-0.36) to remove Illumina-specific adapter sequences and the minimal length was set to 36 (MINLEN) for trimming sequences. The paired-end filtered reads were then aligned to the D. Melanogaster dm6 genome (BDGP Release 6 + ISO1 MT/dm6) by HISAT2 (Galaxy Version 2.2.1 + galaxy0)[65] to generate BAM files with the specific strand information set to "Reverse". Count files were then generated by featureCounts (Galaxy Version 2.0.1 + galaxy2)[66] and the D. Melanogaster reference genome was utilized as the gene annotation file with specific strand information set to "stranded (Reverse)". The resulting count files (tabular format) were then analyzed with DESeq2 (Galaxy Version 2.11.40.7 + galaxy1)[67] with fit-type set to "local", and P-values of less than 0.05 (adjusted for multiple testing) were considered differentially expressed between factor levels. Normalized count reads were outputted for visualization of expression (heatmaps), and Supplementary Data Files 3a contains normalized count reads across all experimental samples. UCSC genome browser visualization: The makeUCSCfile software package from HOMER (v4.11) was utilized to generate bedGraph files for visualizing changes in tag density at exon 2 of clk comparing nCLK-$\Delta1$ and control samples (Supplementary Fig. 2b).

**Heatmap visualizations.** We employed the heatmap2 (Galaxy Version 3.0.1) function from R ggplot2 package to visualize bioinformatics data. Data were not clustered, and data were scaled by row for normalization across timepoints.

**Electroretinogram assays.** ERGs were performed and analyzed in two independent laboratories. ERGs were recorded for eight nCLK-$\Delta1$ female flies reared on AL or DR diets supplemented with vehicle or RU486 at day 14 (18 days of age) at the Baylor College of Medicine (BCM), and at day 21 (25 days of age) at the University of California, Santa Barbara (UCSB). ERGs were recorded for prCLK-$\Delta1$ and prCLK-OE female flies at UCSB reared on AL or DR and maintained at either 18 °C or 30 °C at ages 6, 10, 14, and 18 days old. BCM: ERG recordings were performed as in ref. [68]. Flies were glued on a glass slide. A recording electrode was placed on the eye and a reference electrode was inserted into the back of the fly head. Electrodes were filled with 0.1 M NaCl. During the recording, a 1 s pulse of light stimulation was given. The ERG traces of at least eight flies per genotype/diet were recorded and analyzed by LabChart8 software (AD Instruments). UCSB: Mated female flies (nCLK-$\Delta1$ and prCLK-$\Delta1$) were reared on AL or DR diets starting at 4 days old with and without RU486. ERGs were recorded at ages 6, 10, 14, and 18 days old. ERG recordings were performed as in ref. [69]. Two glass electrodes were filled with Ringer's solution and electrode cream was applied to immobilized flies. A reference electrode was placed on the thorax, while the recording electrode was placed on the eyes. Flies were then exposed to a 10 s pulse of ~200lux white light, a light intensity that is comparable to the phototaxis assay. An EI-210 amplifier (Warner Instruments) was used for amplifying the electrical signal from the eye after light stimulation, and the data were recorded using a Powerlab 4/30 device along with the LabChart 6 software (AD Instruments). Raw data were then uploaded into R-statistical software for plotting and statistical analysis. All electroretinograms were performed between ZT4-8 or ZT12-14.

**Positive phototaxis assay.** Positive phototaxis was performed using an adapted protocol from ref. [70]. Phototaxis measurements were recorded longitudinally on populations of female flies aged (6, 10, 14, 18, and 22 days old) on either AL or DR

food (with or without 200 μM RU486 when indicated) at a density of 10–25 flies per tube prior to and after phototaxis measurements. Approximately 160–480 flies were used in each phototaxis experiment. On the day of phototaxis recording, eight groups of flies (four AL and four DR groups) were placed in separate 2.5 × 20 cm tubes (created from three enjoined narrow fly vials [Genesee Scientific]) and dark-adapted for 15 min prior to light exposure (no food was available in the tubes during phototaxis assays). Flies were then gently tapped to the bottom of the tube, placed horizontally, and exposed to white light from an LED strip (Ustellar, UT33301-DW-NF). A gradient of light intensity was created, with 500lux at the nearest point in the fly tube to the light source and 150lux at the furthest point. Phototaxis activity was recorded by video at 4 K resolution (GoPro, Hero5 black). Positive phototaxis was scored manually as the percentage of flies that had traveled >19 cm toward the light source in three 15 s intervals (15 s, 30 s, and 45 s). "Phototaxis index" was calculated by averaging the percent of positive phototaxis for each vial at the three 15 s intervals. To control for light-independent wandering activity, a phototaxis index was also calculated when the light source was placed in parallel to the fly tube, such that all parts of the tube were equally illuminated with 500lux. We accessed positive phototaxis behavior from the following fly strains in this study: Canton-S, clk$^{out}$, cry01, cry02, cryB, nCLK-Δ1, nCLK-Δ2, GMR-GAL4 > mCherry-RNAi (cnt), GMR-GAL4 > arr1-RNAi, GMR-GAL4 > ATPα-RNAi, GMR-GAL4 > retinin-RNAi, GMR-GAL4 > sun-RNAi, GMR-GAL4 > nrv2-RNAi, GMR-GAL4 > nrv3-RNAi, prCLK-Δ1, prCLK-OE, Spa-GAL4 > ATPα-RNAi, and Spa-GAL4 > RNAi-cnt.

**RNA extraction and cDNA preparation**. Adult female flies (strains nCLK-Δ1, nCLK-Δ2, w$^{1118}$; ninaE$^{17}$, rh3$^2$, rh4$^1$, rh6$^G$, GMR-GAL4 > ATPα-RNAi, GMR-GAL4 > mCherry-RNAi) were maintained on AL or DR for, then flash-frozen on dry ice (11 days old). Heads were separated from bodies (thorax and abdomen) by vigorous shaking. Flies were then ground using a hand-held homogenizer at room temperature following MiniPrep instructions. Total RNA was isolated using the Quick-RNA MiniPrep Kit (Zymo Research, 11-328). RNA was collected into 30 μl DNAse/RNAse-free water and quantified using the NanoDrop 1000 Spectrophotometer (Thermo Scientific). For each experiment, 120–180 age-, genotype-, and diet-matched flies were collected, and three independent RNA extractions were performed. To extract RNA from heads, 40–60 flies were used; to extract RNA from bodies, 20–30 flies were used. cDNA preparation: The iScript Reverse Transcription Supermix for RT-qPCR (Bio-Rad, 1708841) was used to generate cDNA from RNA extracted from heads and bodies. For each group, 1 μg of total RNA was placed in a volume of 4 μl iScript master mix, then brought to 20 μl with DNAse/RNAse-free water. A T1000 thermocycler (BioRad) was used for first-strand RT-PCR reaction following iScript manufacturers' instructions—priming step (5 min at 25 °C), reverse transcription (30 min at 42 °C), and inactivation of the reaction (5 min at 85 °C).

**Real-time quantitative PCR**. Reactions were performed in a 384-well plate. Each reaction contained 2 μl of 1:20 diluted cDNA, 1 μl of primers (forward and reverse at 10 μM), 5 μl SensiFAST SYBR Green No-ROX Kit (BIOLINE, BIO-98020), and 2 μl of DNAse/RNAse-free water. The qPCR reactions were performed with a Light Cycler 480 Real-Time PCR machine (Roche Applied Science) with the following run protocol: pre-incubation (95 °C for 2 min), forty PCR cycles of denaturing (95 °C for 5 s, ramp rate 4.8 °C/s), and annealing and extension (60 °C for 20 s, ramp rate 2.5 °C/s). The PCR primer sequences (forward and reverse) are in Supplementary Data 7.

**Hemolymph mass spectrometry**. Proteomic sample preparation: nCLK-Δ1 female flies (Elav-GeneSwitch-GAL4 > UAS-nCLK-Δ1) were reared on AL diet plus RU486 or vehicle control (N = 300 flies per biological replicate, n = 3 biological replicates). On day 14 (18 days old), flies were snap-frozen on dry ice and transferred to prechilled vials. The vials were vortexed for 5–10 s to remove heads and the frozen bodies were transferred to room temperature vials fitted with 40 μm filters. Headless bodies were thawed at room temperature for 5 min and spun at 2000 × g for 10 min at 4 °C. Following the spin, hemolymph collected at the bottom of each vial, and the bodies remained within the filters. Digestion: A Bicinchoninic Acid protein assay (BCA) was performed for each of the hemolymph samples and a 100 μg aliquot was used for tryptic digestion for each of the six samples. Protein samples were added to a lysis buffer containing a final concentration of 5% SDS and 50 mM triethylammonium bicarbonate (TEAB), pH ~7.55. The samples were reduced to 20 mM dithiothreitol (DTT) for 10 min at 50 °C, subsequently cooled to room temperature for 10 min, and then alkylated with 40 mM iodoacetamide (IAA) for 30 min at room temperature in the dark. Samples were acidified with a final concentration of 1.2% phosphoric acid, resulting in a visible protein colloid. 90% methanol in 100 mM TEAB was added at a volume of seven times the acidified lysate volume. Samples were vortexed until the protein colloid was thoroughly dissolved in the 90% methanol. The entire volume of the samples was spun through the micro S-Trap columns (Protifi) in a flow-through Eppendorf tube. Samples were spun through in 200 μL aliquots for 20 s at 4000 × g. Subsequently, the S-Trap columns were washed with 200 μL of 90% methanol in 100 mM TEAB (pH ~7.1) twice for 20 s each at 4000 × g. S-Trap columns were placed in a clean elution tube and incubated for 1 h at 47 °C with 125 μL of trypsin digestion buffer (50 mM

TEAB, pH ~8) at a 1:25 ratio (protease:protein, wt:wt). The same mixture of trypsin digestion buffer was added again for overnight incubation at 37 °C.

Peptides were eluted from the S-Trap column the following morning in the same elution tube as follows: 80 μL of 50 mM TEAB was spun through for 1 min at 1000 × g. Eighty microliters of 0.5% formic acid was spun through next for 1 min at 1000 × g. Finally, 80 μL of 50% acetonitrile in 0.5% formic acid was spun through the S-Trap column for 1 min at 4000 × g. These pooled elution solutions were dried in a speed vac and then resuspended in 0.2% formic acid. Desalting: The resuspended peptide samples were desalted with stage tips containing a C18 disk, concentrated, and resuspended in aqueous 0.2% formic acid containing "Hyper Reaction Monitoring" indexed retention time peptide standards (iRT, Biognosys). Mass-spectrometry system: Briefly, samples were analyzed by reverse-phase HPLC-ESI-MS/MS using an Eksigent Ultra Plus nano-LC 2D HPLC system (Dublin, CA) with a cHiPLC system (Eksigent) which was directly connected to a quadrupole time-of-flight (QqTOF) TripleTOF 6600 mass spectrometer (SCIEX, Concord, CAN). After injection, peptide mixtures were loaded onto a C18 precolumn chip (200 μm × 0.4 mm ChromXP C18-CL chip, 3 μm, 120 Å, SCIEX) and washed at 2 μl/min for 10 min with the loading solvent (H$_2$O/0.1% formic acid) for desalting. Subsequently, peptides were transferred to the 75 μm × 15 cm ChromXP C18-CL chip, 3 μm, 120 Å, (SCIEX), and eluted at a flow rate of 300 nL/min with a 3 h gradient using aqueous and acetonitrile solvent buffers. Data-dependent acquisitions (for spectral library building): For peptide and protein identifications the mass spectrometer was operated in data-dependent acquisition[53] mode, where the 30 most abundant precursor ions from the survey MS1 scan (250 msec) were isolated at 1 m/z resolution for collision-induced dissociation tandem mass spectrometry (CID-MS/MS, 100 msec per MS/MS, 'high sensitivity' product ion scan mode) using the Analyst 1.7 (build 96) software with a total cycle time of 3.3 s as previously described[71]. Data-independent acquisitions: For quantification, all peptide samples were analyzed by data-independent acquisition (DIA, e.g., SWATH, SWAG), using 64 variable-width isolation windows[72,73]. The variable window width is adjusted according to the complexity of the typical MS1 ion current observed within a certain m/z range using a DIA 'variable window method' algorithm (more narrow windows were chosen in 'busy' m/z ranges, wide windows in m/z ranges with few eluting precursor ions). DIA acquisitions produce complex MS/MS spectra, which are a composite of all the analytes within each selected Q1 m/z window. The DIA cycle time of 3.2 s included a 250 msec precursor ion scan followed by 45 msec accumulation time for each of the 64 variable SWATH segments.

**Identification of photoreceptor-enriched CLK-output genes**. Diagram of bioinformatics steps reported in Supplementary Fig. 5A. Gene lists are reported in Supplementary Data 11. We identified the top 1000 photoreceptor-enriched genes from ref. [74] (GSE93782). We then filtered this list for genes that oscillate in a circadian fashion, and that are downregulated with age from ref. [16] (GSE81100). Approximately 1/3 of the photoreceptor-enriched genes (366 genes) were expressed in a circadian fashion in young wild-type heads and approximately one-half of these (172 genes) displayed a significant loss in expression with age (5- vs 55-day old heads). We further analyzed the remaining gene lists to identify those that are significantly upregulated on DR compared to AL at either ZT 0 or ZT 12 from control (vehicle-treated) samples from our nCLK-Δ1 RNA-Seq analyses. Transcripts with a DESeq2 $P \leq 0.05$ (non-adjusted) were considered differentially expressed. To increase the chance of including false negatives we utilized the raw P-values instead of the adjusted P-values (multiple testing correction) and performed additional experiments to validate these downstream targets. For the final filtering step, we analyzed the genes that were significantly downregulated in nCLK-Δ1 on DR (RU486 vs vehicle-treated controls), resulting in the identification of Gβ76c, retinin, and sun.

**Tangential retinal sectioning and imaging**. Control, prCLK-OE, and prCLK-Δ1 female flies were reared on AL or DR food for 2 or 10 days. Flies were then decapitated (6 and 14 days old) and whole-fly heads were fixed in 2.5% glutar-aldehyde overnight and then transferred into 2.0% osminium tetraoxide for approximately 4 h. The heads were then dehydrated in 100% ethanol and embedded in Epon. Tangential retinal sections (~0.5 micron slices) were stained with 0.1% Toluine Blue. Image capture was performed with a Nikon Ni-E upright microscope with a motorized stage, MQA18000 DS-Fi3 Microscope Camera controlled by the NIS Elements 5.20 software. The microscope was set to 'brightfield', auto-exposure was set to 'continuous', power at 100%, and auto-white was selected for each image taken with either a ×20 and ×40 objective.

**Identification of differential gene expression in flies reared in LD vs DD**. Gene expression changes in heads of y.w. flies housed in 12:12 LD vs constant darkness (DD) were accessed from ref. (GSE3842)[13]. Fold-changes in response to light were calculated by averaging the changes in expression at each timepoint from a circadian time-course microarray and comparing expression between flies housed in LD vs. DD. Individual genes were scored as significantly differentially expressed by performing a Student's t-test (paired, two-tailed) with $P \leq 0.05$.

**Statistics and reproducibility**. The individual biological replicates "n" and the number of individual flies "N" is denoted in each figure legend along with the particular statistical test utilized. The P-value statistics are included in each figure. All error bars are represented as the standard error of the mean (SEM), and all graphs were generated in PRISM 9 (GraphPad). The experiments in this manuscript were performed with populations of female flies (i.e., typically greater than 20 flies per biological replicate).

Figure 3b: The number of biologically independent flies used are as follows (Day 2, 6, 10, 14): AL cnt 10, 14, 14, 10 flies, AL prCLK-Δ1 4, 5, 4, 0 flies, AL prCLK-OE 10, 10, 9, 10 flies, DR cnt 10, 10, 10, 10 flies, DR prCLK-Δ1 5, 6, 7, 0 flies, DR prCLK-OE 10, 10, 12, 10 flies. The ERG amplitudes for the prCLK-Δ1 flies on day 14 were non-responsive (flat) so we did not include those data in the graph. The ERG amplitudes were collected from flies examined over two independent experiments.

Figure 3c: The number of biologically independent flies with similar observable phenotypes to the main figure are as follows (Day 2, 10): AL cnt 4, 2 flies, AL prCLK-Δ1 2, 3 flies, AL prCLK-OE 2, 2 flies, DR cnt 3, 6 flies, DR prCLK-Δ1 3, 1 flies, DR prCLK-OE 4, 3 flies.

Figure 5: The number of flies used for lifespan analyses are as follows: (a) LD housed flies: AL $N = 560$, DR $N = 584$; DD housed flies: AL $N = 460$, DR $N = 462$. (b-g) $w^{1118}$; $ninaE^{17}$ flies: AL $N = 514$, DR $N = 511$; $w^{1118}$; $rh3^2$ flies: AL $N = 543$, DR $N = 597$; $w^{1118}$; $rh4^1$ flies: AL $N = 550$, DR $N = 593$; $w^{1118}$; $rh6^G$ flies: AL $N = 533$, DR $N = 563$; $w^{1118}$; $Gq\alpha^1$ flies: AL $N = 403$, DR $N = 400$. (h) RNAi control flies: AL $N = 365$, DR $N = 351$; $arr1$-RNAi flies: AL $N = 333$, DR $N = 322$. (i) Retinal-treated flies: AL $N = 289$, DR $N = 236$; Vehicle-treated flies: AL $N = 256$, DR $N = 126$. (j) RNAi control flies: AL $N = 493$, DR $N = 490$; $ATP\alpha$ RNAi flies: AL $N = 510$, DR $N = 535$. Survival data is plotted as an average of three independent lifespan repeats.

**Time-course microarray analyses**. Four independent biological replicates (per diet/timepoint) of approximately 35 *Canton-S* female flies were collected on dry ice every 4 h for 20 h starting at ZT 0 (six total timepoints, 48 total samples). Differential expression was determined by two-tailed Student's *t*-test (paired) comparing the averaged transcript-level expression values between AL and DR samples across all timepoints, and P-values less than 0.05 were considered significant. The JTK_CYCLE algorithm[75] (v3.0) was utilized to identify circadian transcripts from the AL and DR time-course expression arrays. Transcript-level expression values for each of the four biological replicates (per timepoint/diet) were used as input for JTK_CYCLE, and the period length was set to 24 h. We defined circadian transcripts as those displaying a JTK_CYCLE P-value of less than 0.05 (non-adjusted) in wild-type flies (*Canton-S*) while displaying a JTK_CYCLE P-value of greater than 0.05 (non-adjusted) in circadian mutant flies ($tim^{01}$). Subsequent analyses compared diet-dependent changes in JTK_CYCLE outputs (phase and amplitude).

**nCLK-Δ1 RNA-Seq**. Three independent biological replicates of 100 mated female adult flies were utilized per genotype/diet/timepoint. DEseq2 software[67] was utilized and P-values of less than 0.05 (adjusted for multiple testing) were considered differentially expressed between factor levels for gene-ontology enrichment analyses.

**ERG responses**. For ERG experiments we quantified responses from 6 to 15 individual flies per standard in the field. Statistical significance was determined by two-tailed Student's *t*-test (unpaired), comparing ERG responses between diet and genotypes. Full ERG statistics are reported in Supplementary Data 6.

**Survival analyses**. The Log-rank (Mantel-Cox) test was used to determine statistical significance by comparing average lifespan curves from a minimum of two independent lifespan replicates. Hazard Ratios (log-rank) were also utilized to determine the probability of death across genotypes, lighting conditions, and diet. Detailed Log-rank and hazard ratios for each lifespan are reported in Supplementary Data 10.

**Positive phototaxis assay**. Statistical significance for the phototaxis index at each timepoint was calculated with the Student's *t*-test (two-tailed, unpaired). Two-way ANOVA or mixed-effects models were performed to determine statistical significance between diet, genotype, or time interactions. Full statistical output (two-way ANOVA and *t*-test) for all phototaxis experiments are reported in Supplementary Data 5.

**Real-time quantitative PCR**. Fold-change in gene expression was calculated using the ΔΔCt method and the values were normalized using *rp49* as an internal control. P-values were calculated with the pairwise Student's *t*-test comparing Log2 fold-changes in expression.

**Mass-spectrometric data processing, quantification, and bioinformatics**. Mass-spectrometric data-dependent acquisitions[53] were analyzed using the database search engine ProteinPilot (SCIEX 5.0 revision 4769) using the Paragon algorithm (5.0.0.0,4767). Using these database search engine results a MS/MS spectral library was generated in Spectronaut 14.2.200619.47784 (Biognosys). The DIA/SWATH data was processed for relative quantification by comparing peptide peak areas from various different timepoints during the cell cycle. For the DIA/SWATH MS2 data sets quantification was based on XICs of 6-10 MS/MS fragment ions, typically y- and b-ions, matching to specific peptides present in the spectral libraries. Differential protein expression analysis was performed using a paired *t*-test and P-values were adjusted for multiple testing. Peptides were identified at $Q \leq 0.01\%$, and significantly changed proteins were accepted at a 5% FDR ($Q \leq 0.01$).

**Gene-ontology enrichment analysis**. To identify enriched gene-ontology (i.e., bioprocess) categories with the resultant lists from bioinformatics approaches, we utilized the "findGO.pl" package from HOMER (v4.11). Full gene-ontology lists including enrichment statistics (calculated assuming hypergeometric distribution, no adjustment for multiple-hypothesis testing) and associated gene lists are reported in supplementary data files. A maximal limit of 200 gene identifiers per GO category was implemented to reduce the occurrence of large, over-represented terms that lack specificity (i.e., metabolism).

**Reporting summary**. Further information on research design is available in the Nature Research Reporting Summary linked to this article.

## Data availability

The time-course microarray data and accompanied JTK_CYCLE statistics that support the findings in this study have been deposited in the Gene Expression Omnibus[76] with the GSE158286 accession code The RNA-seq data and accompanied differential expression analyses that support the findings in this study have been deposited to GEO with the GSE158905 accession code . The mass-spectrometric raw data generated in this study have been deposited at with the MassIVE ID MSV000086781; it is also available at ProteomeXchange with the ID PXD023896. Additional mass-spectrometric details from DIA and DDA acquisitions, such as protein identification and quantification details are available at the repositories (including all generated Spectronaut and Protein Pilot search engine files). The wild-type circadian gene expression data analyzed in this study are available in the GEO database under accession code GSE81100[16]. The LD vs DD gene expression data analyzed in this study are available in the GEO database under accession code GSE3842[13]. Data on photoreceptor-enriched genes analyzed in this study are available in the GEO database under accession code GSE93782[74]. All other data supporting the findings of this study are included within the manuscript and supplementary information/data files. Source data are provided with this paper.

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

## Acknowledgements
We would like to thank the QB3 Genomics Functional Genomics Lab at UC Berkeley and the Vincent J. Coates Genomics Sequencing Lab for RNA-seq library preparation and deep sequencing. We would like to acknowledge Hugo Bellen and Zhongyuan Zuo for contributing resources and insight. We thank Daron Yim and John McCarthy for their assistance. This work was supported by the American Federation of Aging Research, NIH grants R01AG038688 and AG045835 and the Larry L. Hillblom Foundation to P.K., NIH/ NIA T32 award AG000266 to B.A.H., NIH F31EY033179-01 to G.T.M., Larry L. Hillblom Foundation 2019-A-02-FEL to S.B., NIH grants R01-EY008117, R01-AI169386, and R01-DC007864 to C.M. We acknowledge the Buck Institute Proteomics Core and the support of instrumentation from the NCRR shared instrumentation grant 1S10 OD016281 to B.S.

## Author contributions
Conceptualization, B.A.H., G.T.M., and S.K; Software, B.A.H., G.T.M., B.S., and S.K; Formal analysis, B.A.H., G.T.M., and B.S.; Investigation, B.A.H., G.T.M., C.L., T.L., N.L., D.L.-K., and S.B.; Visualization, B.A.H.; Writing—original draft, B.A.H., Writing—review and editing, B.A.H., G.T.M., and P.K.; Data curation, B.A.H., G.T.M., S.M., and B.S.; Methodology, G.T.M. and M.L.; Resources, C.M. and P.K.; Supervision, P.K.; Funding acquisition, C.M. and P.K.

## Competing interests
The authors declare no competing interests.
