## [Peer Review File · Nature Communications]

REVIEWER COMMENTS

Reviewer #1 (Remarks to the Author):

Previous studies have shown a two-ways interaction between dietary restriction (DR) and the circadian clocks. In flies, the DR-induced lifespan extension is decreased when peripheral (non-brain) circadian clocks are stopped and DR increases the amplitude of clock protein cycling (Katewa et al. 2016). Building on these data Hodge et al. investigate the effects of clock disruption in the eye and retinal degeneration on lifespan in DR or ad libitum feeding conditions. They conclude that flies expressing the dominant-negative CLOCK protein show accelerated retinal senescence and shortened lifespan. They show that photoreceptor activation and subsequent retinal senescence shorten lifespan and that DR protects against this effect.

The idea that photoreception activation shortens lifespan and is prevented by DR is very interesting and the possibility that the eye circadian clock protects from these effects exciting. The conclusions of this study are definitely important. The data are rather convincing although I have some concerns with the tools that are used to disrupt the clock and I would recommend using classical clock mutants to confirm the results, as detailed below.

1- Since CLK affects the anatomy of at least some clock neurons, the CLK-delta dominant-negative might have related effects in other cells including photoreceptors. It would thus be important to use an independent way of abolishing the clock function. *per0* or *tim0* flies do not have reported developmental effects, including in the eye. I think that the authors need to confirm the CLK-delta experiments for at least the phenotypes phototaxis assay during aging in AL and DR conditions, to conclude that the absence of a functional clock prevents

2- It is clear from the present study that CLK-delta flies have a strongly reduced lifespan in both AL and DR conditions and it is not clear to me that it is the case of *per0* flies. As indicated for photoreceptor degeneration, *per0* or *tim0* flies need to be used in the lifespan assay in AL and DR to decipher between the effects of expressing the CLK-delta protein and the absence of a functional clock.

3- p17. The three CLK-controlled genes (*Gb76c*, *retinin*, *sunglasses*) whose downregulation causes phototaxis defects and shortening of lifespan under DR (Fig 5e-f) should show circadian oscillations of their mRNAs with larger amplitude in DR versus AL conditions if the proposed model is correct. Is it the case ?

4- I notice some tendency to mix known results from mammals and flies to draw general conclusions. I would suggest to be more cautious about this since rather strong differences exist, for example in the phototransduction pathway. Please be more precise about the model system when citing published data and provide appropriate references.

Minor points

- p8. "These findings provide a potential mechanistic explanation for the rhythmic response pattern in light-sensitivity observed in *Drosophila* photoreceptors". It is well documented in vertebrates but is it

clearly shown in flies? Please provide references.

Reviewer #2 (Remarks to the Author):

The manuscript by Hodge and colleagues demonstrates that the circadian clock regulates phototransduction in such a manner as to increase lifespan, visual function, and lower systematic inflammation. The data in the paper are convincing and support the conclusion of the authors.

I am having some trouble with the connections that are made in the paper. For example, it is not clear to me why photoreceptor degeneration would cause an increase in systemic inflammation and a decrease in lifespan. I am not questioning the data in the paper. I just think that the authors need to connect the dots for the reader more carefully.

I am also not sure I understand how the dietary restriction plays into the study. I think the authors could just look at defective retinal clocks and the effect that it has on the fly. Especially since flies are not normally going to encounter dietary restrictive conditions out in the wild.

I also think that the authors need to focus on one of the effects that the eye clock has on the fly. It could be visual function, inflammation, or lifespan but I think the paper will benefit from a single idea.

Just my two cents - I think the strongest part of the paper is the effect the eye clock has on phototransduction and visual function. I think that a revised paper in which the authors focus exclusively on this connection will be very exciting and would be appropriate for Nature Communications.

The effect on systematic inflammation and lifespan, while both interesting, are hard to connect to the state of photoreceptor health and, to me, make a much less interesting story.

If the authors focus on visual function I would suggest that they combine their clock mutants with all known phototransduction mutants to gain a comprehensive view of how the clock is affecting degeneration and visual function.

I would also suggest that the authors do experiments in complete darkness, a light:dark cycle, and constant light.

Reviewer #3 (Remarks to the Author):

Using *Drosophila* models, in this study Hodge et al. demonstrated that dietary restriction (DR) extends lifespan by promoting circadian homeostatic processes that protect the visual system from age- and light-associated damage. The authors have further demonstrated that disrupting circadian rhythms in the eye by inhibiting the transcription factor, Clock (CLK), or CLK-output genes, accelerated visual

senescence, induced a systemic immune response, and shortened lifespan. Moreover, flies subjected to dietary restriction were protected from the lifespan-shortening effects of photoreceptor activation. Inversely, photoreceptor inactivation, achieved via mutating rhodopsin or housing flies in constant darkness, primarily extended lifespan in flies reared on a high-nutrient diet. Overall, their findings establish the eye as a diet-sensitive modulator of lifespan and indicate that vision is an antagonistically pleiotropic process that contributes to organismal aging.

Using multiple assays this is a very detail, systemic, and well-designed mechanistic study to demonstrate the linkage to DR and circadian rhythms to protect the visual system from age- and light-associated damage. Overall, findings from this study are quite interesting, however, require more clarity, specification, and justifications. In my assessment with some clarity and justification, this might become suitable for publication in Nature Communications. The following questions/concerns need to be addressed for this consideration.

1. The authors have used multiple drivers to express/knockdown various genes in multiple tissues, however, the focus is limited to visual-aging. The authors need to provide a table using all the drivers/genes used in this manuscript for all the genetic modulations with clear/concise justifications. This is essential to justify the authors' hypothesis and potential outcome. This will be also useful for the general audience in understanding the power of *Drosophila* genetics for addressing cell-autonomous functions.
2. Related to my concern 1, the authors frequently use ELAV and GMR drivers however, did not differentiate clock-mediated neuroprotective functions vs photoreceptors/visual system. When using a Pan-neuronal driver like ELAV, how modulation of circadian clock or DR affecting the neuronal system? Based upon their findings, it is clear that the life-span and phototaxis performance will be affected by the para-neuronal ELAV driver. Therefore, most of the outcomes shown by the authors could be neuronal as well. Therefore, it will be essential to differentiate neuronal vs visual systems in the entire manuscript. Also, as shown in Fig.5, it is important to clarify how knock-down of GMR-driven DR-sensitive eye-specific Clk-output genes reduces survival and affects phototaxis performance? Recent studies have shown that GMR-expression is not limited to the visual system therefore, most of these genes knock-down, including reduces survival and affects phototaxis performance might be not limited to the visual system.
3. It is well known that GMR-driver mediated expression/knock-down directly affects ommatidia organization/degeneration. Did the authors see any defects in ommatidia organization/degeneration with the ocular expression of CLK-delta1 and CLK- delta or ocular knock-down of ATP or other genes shown in listed in Fig. 5? It will be also important to know if this was impacted by DR or other genetic modulations.
4. As shown in Fig. 3, NCLK-1 flies display elevated immune responses and shortened lifespan and demonstrated that disrupting neuronal CLK function elevates systemic immune responses. Later the authors have demonstrated that this is also eye-driven (Fig. 3c, d). For these experiments, the authors have used fly bodies, instead of heads. Why expression was not tested in the head/eyes?
5. Similarly, in Fig 5e, why the experiment is limited to the whole fly? Relative mRNA expression of immune genes (AttaA, DiptB, and Dro) calculated by RT-qPCR with mRNA isolated from bodies of w1118 and rhodopsin mutant flies housed in 12:12h LD.
6. As I mentioned before, different drivers used for different experiments, however, the conclusion is limited to eye aging? The authors need to justify how the expression of different tissues leads to eyes aging? A clear cell-autonomous vs non-cell-autonomous justification will be required including editing

Fig. 6. If neuronal damage signals propagate throughout the body to drive systemic immune responses, the authors need to find basal expression of these genes responsible for immune response in the neuronal and visual tissue/system.

7. Again, the authors have argued that neuronal CLK function is required for the full lifespan extension mediated by DR and indicate that photoreceptor clocks are essential for the maintenance of visual function with age and organismal survival? I think these are confusing statements and need more clarity and justification. Despite several pieces of evidence provided by authors for the neurological role, I am not sure how the outcome is limited to visual function/system?

8. DR-protection against lifespan shortening downstream of light and/or rhodopsin-mediated signaling in a manner that requires light-adaptation, and by extension, arr1-mediated rhodopsin endocytosis. This justification needs more clarification.

9. Despite the noticeable difference of immune system and photoreceptors cells between, *Drosophila* with the mammalian visual system, the authors did not indicate the limitations of their finding.

10. The discussion section of the main text needs a significant improvement including clarity about the above-mentioned comments. The authors have also used discussion/significance of their findings in the SI section without any clarification.

Dear Editor,

Thank you for considering our manuscript in your esteemed journal and allowing us to respond to the reviewer's comments. We thank the reviewers for their thoughtful comments and I hope you will find that our responses have significantly improved the manuscript. We have made appropriate changes in the text and figures and here is our point-by-point response to the comments.

We have provided the following new data and updates to address the reviewer comments:

New Figures/Data:

Figure	Data
3a-c	prCLK-DN1 and prCLK-OE phototaxis, ERG, transgenial cross-sections.
S. 1b, d-f, l	Tim01 circadian transcriptome analyses.
S. 4b	prCLK-DN1 and prCLK-OE ERG traces.
S. 7c	arr1 expression from nCLK-DN1 RNA-Seq.
S. 8a-b	Cone-cell specific ATPalpha-RNAi knockdown phototaxis and lifespan.
S. Data 12	Rationale and strengths/weakness of lines used in study.

Updated Figures/Data:

Figure	Data
4f	prCLK-DN1 lifespan (N=3).
S. 7d	Eye-specific arr1 knockdown phototaxis.
S. Data 1	AL and DR circadian transcriptome analyses in wild-type and circadian mutant flies
S. Data 7	Survival analyses.
S. Data 9	Positive phototaxis responses and statistics.
S. Data 10	Electroretinogram analyses and statistics.
S. Table 2	Drosophila strains used in this study.

Reviewer #1 (Remarks to the Author):

Previous studies have shown a two-ways interaction between dietary restriction (DR) and the circadian clocks. In flies, the DR-induced lifespan extension is decreased when peripheral (non-brain) circadian clocks are stopped and DR increases the amplitude of clock protein cycling (Katewa et al., 2016, PMID: 26626459). Building on these data Hodge et al. investigate the effects of clock disruption in the eye and retinal degeneration on lifespan in DR or ad libitum feeding conditions. They conclude that flies expressing the dominant-negative CLOCK protein show accelerated retinal senescence and shortened lifespan. They show that photoreceptor activation and subsequent retinal senescence shorten lifespan and that DR protects against this effect.

The idea that photoreception activation shortens lifespan and is prevented by DR is very interesting and the possibility that the eye circadian clock protects from these effects exciting. The conclusions of this study are definitely important. The data are rather convincing although I have some concerns with the tools that are used to disrupt the clock and I would recommend using classical clock mutants to confirm the results, as detailed below.

1- Since CLK affects the anatomy of at least some clock neurons, the CLK-delta dominant-negative might have related effects in other cells including photoreceptors. It would thus be important to use an independent way of abolishing the clock function. per0 or tim0 flies do not have reported developmental effects, including in the eye. I think that the authors need to confirm the CLK-delta experiments for at least the phenotypes phototaxis assay during aging in AL and DR conditions, to conclude that the absence of a functional clock prevents

- We agree with the concern that over-expressing CLK-Δ1 might influence the function of non-photoreceptor neurons, especially when we utilizing the pan-neuronal Elav-gene-switch-GAL4 driver system. We would like to point out that with the GeneSwitch system we induce GAL4-mediated expression with the addition of RU486 in 4-day old adult, mated female flies, post-development. Therefore, we can avoid developmental defects associated with the expression of CLK-Δ1. Furthermore, the positive phototaxis and ERG data with the CLK-Δ1 lines, elav-GS-GAL4>CLK-Δ1 and CLK-Δ2, are indistinguishable from control at the earliest time-point recorded (day 2 post induction), indicating that the photoreceptor cells have developed normally. An additional strength of Gene-Switch system is that the control flies are from the same F1 generation and have the same genotype (but are reared on vehicle treated food vs food with RU486). We also note that although we used a pan-neuronal driver (elav-GS-GAL4), Clk is enriched 5-fold within the photoreceptor cells vs the rest of the neurons in the brain (see **Supplemental Fig. 6d**). Furthermore, within the brain, *clk* expression is confined to a small population of approximately 150 neurons. Therefore, it is likely that over-expressing a dominant-negative CLK protein with a pan-neuronal driver may only affect this minor population of neurons in the brain. To demonstrate cell-intrinsic function of CLK we have included new data with photoreceptor-specific disruption to Clk, which we describe in greater detail below.

*Supplemental Fig. 6d

- To directly test the role of CLK function in adult photoreceptors (post-developmental), we combined the *Trpl-Gal4* driver (which narrowly expresses GAL4 in the photoreceptor cells) with temperature-sensitive Gal80 (which strongly represses Gal4 activity at 18°C) and crossed these flies to *UAS-Clk-Δ1* (*Trpl-gal4;gal80>Clk-Δ1*, denoted *prCLK-Δ1*). To suppress the expression of *Clk-Δ1* during development, we housed these flies (*prCLK-Δ1*) at 18 °C. Upon eclosion, we then sorted the flies (*prCLK-Δ1*) onto AL or DR food and then transferred them to 30 °C to de-repress GAL4 and induce the expression of *Clk-Δ1*. To limit confusion, the data generated with *Elav-GS-GAL4* remain in Figure 2, while new data generated with the photoreceptor specific GAL4 line was placed into a new Figure 3. Similar to *Elav-Gs>Clk-Δ1*, *Trpl-gal4;gal80>Clk-Δ1* flies also display accelerated photoreceptor aging, as is apparent via phototaxis (**Fig. 3a**), ERG amplitude (**Fig. 3b**), and tangential cross-sections of the ommatidia (**Fig. 3c**). *Trpl-gal4;gal80>Clk-Δ1* flies also display a lifespan shortening in comparison to their outcrossed controls (*w/+;Trpl-Gal4/+; Gal80/+*) (**Fig. 4f**). This further demonstrates that CLK function within photoreceptors influences lifespan and visual senescence. Lastly, we have included several *Clk* overexpression experiments (*w;Trpl-Gal4/+;UAS-Clk/Gal80*) and demonstrate that increasing the expression of CLK within the photoreceptor cells is sufficient to delay visual senescence and rescue AL-mediated declines, as we observed with phototaxis (**Fig. 3a**), ERG amplitude (**Fig. 3b**), and cross-sections of the ommatidia (**Fig. 3c**). **See Supplemental Fig. 4b for the averaged ERG traces between the control, *prCLK-Δ1*, and *prCLK-OE* lines.

*Fig. 3a-c and Fig. 4f.

- In addition to using the *CLK-Δ1* lines, we have also tested flies homozygous for a separate null *clk* allele, *CLK^{out}*. Consistent with the *CLK-Δ1* data, in *CLK^{out}* flies, we found that that DR failed to delay visual senescence in comparison to AL-fed flies (now in **Supplemental Fig. 3e**).

*Supplemental Fig. 3e

- We agree that testing additional circadian mutants may help support our claim that CLK influences photoreceptor aging and mediates the beneficial effects of DR on the eye. In addition to the wild-type circadian time-course microarray (*Canton-S* flies), we also performed the same time-course in *tim⁰¹* mutant flies (and have now included these data in Supplemental Fig. 1b, d, e-f, and I) **We also deposited the raw and processed files for the *tim⁰¹* circadian time-course microarray at GEO: GSE158286. Interestingly, we found that DR failed to amplify the number of circadian genes in *tim⁰¹* mutants (**Supplemental Fig. 1d**), indicating that normal circadian function is required.

*Supplemental Fig. 1b, d

- For both *per⁰¹* and *tim⁰¹* mutants, the negative feedback loop that represses CLK/CYC heterodimers is disrupted, resulting in low but constitutive transcriptional activity by CLK/CYC (Glossop et al., 1999. PMID:10531060). Because of this caveat—and our model that CLK-output genes mediate DR’s ability to delay eye aging—we decided not to move forward with testing phototaxis in either *per⁰¹* or *tim⁰¹* flies. In agreement with the observation that CLK/CYC activity exists in mutants of the repressive limb of the molecular clock, we found that photoreceptor genes were still upregulated in expression upon DR in *tim⁰¹* flies, although they failed to display a circadian oscillation (**Supplemental Fig. 1I**).

*Supplemental Fig. 1I

- Instead of utilizing *per⁰¹* and *tim⁰¹* mutants (for the reasons described above) we did decide to test the role of cryptochrome (CRY), a core clock component that is a direct light sensor and aids in photoentrainment. We found that three *cry* mutants (*cry^B*, *cry⁰¹*, and *cry⁰²*) all demonstrated clear enhancements in phototaxis on DR compared to AL (see **Supplemental Fig. 3g-i**). This suggests that circadian timing, and/or photoentrainment, may not be as important for mediating the diet-dependent responses in eye aging compared to proper CLK function. Our final reason for not testing *per⁰¹* or *tim⁰¹* mutants is because they harbor null, full-body mutations in *period* and *timeless*.

Therefore, any phototaxis defect we might observe in these flies could be attributable to either developmental abnormalities or the loss of these genes in extra-ocular tissues.

*Supplemental Fig. 3g-i

2- It is clear from the present study that CLK-delta flies have a strongly reduced lifespan in both AL and DR conditions and it is not clear to me that it is the case of *per0* flies. As indicated for photoreceptor degeneration, *per0* or *tim0* flies need to be used in the lifespan assay in AL and DR to decipher between the effects of expressing the CLK-delta protein and the absence of a functional clock.

- We thank you for your concern regarding potential different circadian mutants may yield variable effects on lifespan in the context of DR. As mentioned in the response to your first questions, we believe that neither the *tim⁰¹* nor *per⁰¹* mutant flies serve as a strong model for accessing the absence of functional CLK. Additionally, it is becoming increasingly apparent that disrupting circadian function, or losing CLK activity, can have tissue-specific effects on aging. In turn, lifespan may reflect a summation and/or differences in the individual contributions of aging across cell-types/tissues. For example, loss of *per⁰¹* function in the gut appears to extend lifespan in *Drosophila* (Ulgherait *et al.*, 2020. PMID: 32317636). Our goal in this study was to elucidate the role of CLK function within the photoreceptors and determine its effect on aging and lifespan. Therefore, we feel it is appropriate to confine our efforts to transgenic lines that manipulate CLK activity in a tissue-specific fashion, namely in the photoreceptors and neurons. Also, we would like to highlight that *per⁰¹* mutant lifespans were performed on AL and DR diets in a previous publication from our lab (Subhash *et al.*, 2016. PMID: 26626459) and that *per⁰¹* and *tim⁰¹* failed to display DR-mediated lifespan extension. These results were subsequently refuted by another lab observed that DR extended lifespan in these lines (Ulgherait *et al.*, 2016. PMID: 27916531). It is unclear why there was a discrepancy between the *period* and *timeless* mutant lifespans, but these may be explained by lab-specific differences in lighting conditions (LUX, and LEDs used) or the microbiomes. Again, given that CLK activity can persist in *per⁰¹* and *tim⁰¹* mutants, we believe that these lines are suboptimal for studying CLK's role in modulating the rate of aging of adult photoreceptor cells.

3- p17. The three CLK-controlled genes (*Gbeta76c*, *retinin*, *sunglasses*) whose downregulation causes phototaxis defects and shortening of lifespan under DR (Fig 5e-f) should show circadian oscillations of their mRNAs with larger amplitude in DR versus AL conditions if the proposed model is correct. Is it the case ?

- The CLK-output genes we decided to analyze in this study (*Gbeta76c*, *retinin*, and *sunglasses*) were chosen based on our bioinformatics approach (Supplemental Fig. 9a): they were enriched in photoreceptors (public dataset, Charlton-Perkins *et al.*, 2017. GSE93782), show circadian mRNA expression in young wild-type heads (public dataset, Kuintzle *et al.*, 2017. GSE81100), are down-regulated with age, upregulated on DR vs AL (our data), and are down-regulated in nCLK-Δ1 vs control flies (our data). We did not observe a circadian oscillation (JTK CYCLE p-value<0.05) with *Gbeta76c*, *retinin*, and *sunglasses* in our AL and DR circadian transcriptome analysis, although these genes are all highly rhythmic in wild-type heads (Supplemental Fig. 9b-d). We likely failed to observe a circadian oscillation in these genes in our dataset because our transcriptome analyses incorporated pooled RNA from whole-fly lysates, whereas the circadian RNA sequencing experiment we reference extracted RNA from heads only.

*Supplemental Fig. 9b-d

- Since we performed our nCLK-Δ1 RNA-Seq analysis at ZT 0 and ZT 12 (the peak and trough of their expression, respectively) we were able to observe time-of-day and diet-dependent changes in the expression of *Gbeta76c*, *retinin*, and *sunglasses* from RNA isolated from heads (Supplemental Fig. 9e-g). Our bulk analysis of circadian transcripts that oscillate on both AL and DR revealed that, on average, these transcripts display a more robust amplitude on DR vs AL (Supplementary Fig. 1j). Analysis of our RNA-Seq data from ZT0 and ZT12 indicate that the amplitude (difference in expression fold-change over time) was more robust for *Gbeta76c* (AL: 1.57, DR: 1.77) and *retinin* (AL: 0.93, DR: 1.17), but not for *sunglasses* although the amplitudes were similar (AL: 1.19, DR: 1.15). Although we demonstrate that on average circadian amplitude is elevated on DR vs AL, our model that DR promotes the circadian expression of CLK-output genes does not rely solely on increases in circadian amplitude. It is possible that for several CLK-output genes that are elevated in expression by DR are also more rhythmic in their pattern of oscillation (i.e., their expression more closely fits a sine wave) while not elevating their overall amplitude. Alternatively, it is possible that DR-CLK mediated benefits to the aging photoreceptor are more dependant on overall expression changes to CLK-output genes, rather than their amplitude or rhythmicity.

*Supplemental Fig. 9e-g

4- I notice some tendency to mix known results from mammals and flies to draw general conclusions. I would suggest to be more cautious about this since rather strong differences exist, for example in the phototransduction pathway. Please be more precise about the model system when citing published data and provide appropriate references.

- We thank the reviewer for this concern and would like to highlight that we have included the following section in the Supplemental Discussion 1 section to help clarify potential differences between flies and mammals:

“In mammals, light-activated rhodopsin in rod and cone photoreceptor neurons couples to, and inactivates, cyclic nucleotide gated channels, hyperpolarizing the cell [17]. This is distinctly different from what occurs in the fly, where light-activated rhodopsin couples to a TRP channel, which when activated depolarizes the cell [18]. However, in a third class of mammalian photoreceptors, the intrinsically-photosensitive retinal ganglion cells (ipRGCs), there is a nearly identical mechanism of phototransduction to Drosophila [19]. The ipRGCs play a role in non-image forming light sensation, effecting pupillary constriction and the entrainment of the central circadian clock to light. There is some evidence that eliminating Bmal1 in mice (either specifically in their ipRGCs or throughout their entire body) impairs the functionality of the ipRGCs [20]. This is consistent with what we observed when we disrupted clk in the Drosophila photoreceptors. Together, this suggests that there may be a conserved mechanism through which circadian clocks mediate the health of photoreceptor cells.”

Minor points

- p8. **“These findings provide a potential mechanistic explanation for the rhythmic response pattern in light-sensitivity observed in Drosophila photoreceptors”. It is well documented in vertebrates but is it clearly shown in flies? Please provide references.**

- It appears we accidentally neglected to include the reference within the main text, as it was only referenced in the supplemental discussion (ref #25). We have now included that reference within the main text: Nippe, O.M., et al., Circadian Rhythms in Visual Responsiveness in the Behaviorally Arrhythmic Drosophila Clock Mutant Clk(Jrk). J Biol Rhythms, 2017. 32(6): p. 583-592.

Reviewer #2 (Remarks to the Author):

The manuscript by Hodge and colleagues demonstrates that the circadian clock regulates phototransduction in such a manner as to increase lifespan, visual function, and lower systemic inflammation. The data in the paper are convincing and support the conclusion of the authors.

I am having some trouble with the connections that are made in the paper. For example, it is not clear to me why photoreceptor degeneration would cause an increase in systemic inflammation and a decrease in lifespan. I am not questioning the data in the paper. I just think that the authors need to connect the dots for the reader more carefully.

- We thank the reviewer for their comment. In our current manuscript we believe we provide sufficient evidence that the photoreceptor is both a modulator of local and systemic inflammation as well as *Drosophila* lifespan. We believe that further studies are needed to elucidate the actual mechanism(s) by which photoreceptor's influence the health of other cell-types and modulate lifespan, but we feel that these studies are not within the scope of this current study. Herein, our focus was to identify the downstream CLK-controlled processes that are upregulated on DR, and to determine how they regulate aging and longevity, which we believe we have provided sufficient evidence.

I am also not sure I understand how the dietary restriction plays into the study. I think the authors could just look at defective retinal clocks and the effect that it has on the fly. Especially since flies are not normally going to encounter dietary restrictive conditions out in the wild.

- Given the observations that circadian clocks decline with age and that restrictive diets such as dietary restriction or calorie restriction enhance core molecular clock rhythms, we set out to identify which processes and tissues displayed clear DR-dependent improvements in circadian oscillation. Our time-course circadian transcriptome data on DR and AL led us to identify light-response genes as circadian and being selectively amplified by DR. We do believe that circadian clocks play important roles even in standard diets, or those that would be consumed in the wild, but our goal was to elucidate the circadian processes downstream of DR that help to slow aging and extend lifespan.

I also think that the authors need to focus on one of the effects that the eye clock has on the fly. It could be visual function, inflammation, or lifespan but I think the paper will benefit from a single idea.

- We and the editor disagree with the critique that the manuscript will benefit by changing the focus or scope, and therefore we will not comment here.

Just my two cents - I think the strongest part of the paper is the effect the eye clock has on phototransduction and visual function. I think that a revised paper in which the authors focus exclusively on this connection will be very exciting and would be appropriate for Nature Communications.

- Please refer to our previous statement.

The effect on systemic inflammation and lifespan, while both interesting, are hard to connect to the state of photoreceptor health and, to me, make a much less interesting story.

- We found through an unbiased analysis of the genes that were upregulated with loss of CLK function that there were significant enrichments in inflammatory genes (i.e., AMPs). Furthermore, housing flies in LD vs DD also increases the expression of inflammatory markers. We then demonstrate that forcing photoreceptor degeneration (eye-specific ATPalpha knockdown) was sufficient to drive a systemic inflammatory response, while reducing rhodopsin mediated signaling (Rhodopsin null lines) suppressed this response. Given that chronic inflammation is now considered a hallmark of aging, we believe it is important to demonstrate that the photoreceptor could be a critical regulator of inflammation in the fly, and although the lifespan and inflammation findings are highly correlated, we are careful to refrain from claiming that photoreceptor derived inflammation is a main driver of lifespan shortening. We believe future studies are needed to develop a likely complex, mechanistic understanding of the interplay of diet, circadian rhythms, photoreceptor physiology, inflammation, and lifespan.

If the authors focus on visual function I would suggest that they combine their clock mutants with all known phototransduction mutants to gain a comprehensive view of how the clock is affecting degeneration and visual function.

- We thank the reviewer for this suggestion. Although we agree that a deeper understanding of how functionally circadian clocks may influence photoreceptor degeneration in varying phototransduction mutants, we believe that these experiments would not change the overall interpretations made in this manuscript and are therefore out of our current scope. Additionally, there are many phototransduction mutants (n>25 different lines) and there is not a clear rationale as to why we should screen these different mutants in the context of our circadian mutants.

I would also suggest that the authors do experiments in complete darkness, a light:dark cycle, and constant light.

- We have performed lifespans in flies in either LD vs DD and demonstrated that the w^{1118} flies display a clear diet-dependent lifespan shortening in LD (Fig. 5a) that is rescued in DD. Alternatively, we found that red-eyed *Canton-S* flies did not display a clear lifespan shortening in LD (Supp. Fig. 7a). We reasoned since w^{1118} flies lack red-pigment in their pigment cells, they are known to be more susceptible to light-mediated photoreceptor degeneration that is likely exacerbated in AL. We also performed the LD vs DD lifespan analysis in flies that lack photoreceptors (TRP^{P365}) and demonstrate that these flies are not long-lived in DD (Supp. Fig. 7b). Our lab previously demonstrated that wildtype *Canton-S* flies are short-lived when reared in constant light vs LD (Katewa *et al.*, 2016, PMID: 26626459), and therefore did not feel the need to repeat these experiments in the current manuscript.

*Fig. 5a

*Supplemental Fig. 7a-b

*Katewa *et al.*, 2016

Using *Drosophila* models, in this study Hodge et al. demonstrated that dietary restriction (DR) extends lifespan by promoting circadian homeostatic processes that protect the visual system from age- and light-associated damage. The authors have further demonstrated that disrupting circadian rhythms in the eye by inhibiting the transcription factor, Clock (CLK), or CLK-output genes, accelerated visual senescence, induced a systemic immune response, and shortened lifespan. Moreover, flies subjected to dietary restriction were protected from the lifespan-shortening effects of photoreceptor activation. Inversely, photoreceptor inactivation, achieved via mutating rhodopsin or housing flies in constant darkness, primarily extended lifespan in flies reared on a high-nutrient diet. Overall, their findings establish the eye as a diet-sensitive modulator of lifespan and indicate that vision is an antagonistically pleiotropic process that contributes to organismal aging.

Using multiple assays this is a very detail, systemic, and well-designed mechanistic study to demonstrate the linkage to DR and circadian rhythms to protect the visual system from age- and light-associated damage. Overall, findings from this study are quite interesting, however, require more clarity, specification, and justifications. In my assessment with some clarity and justification, this might become suitable for publication in Nature Communications. The following questions/concerns need to be addressed for this consideration.

1. The authors have used multiple drivers to express/knockdown various genes in multiple tissues, however, the focus is limited to visual-aging. The authors need to provide a table using all the drivers/genes used in this manuscript for all the genetic modulations with clear/concise justifications. This is essential to justify the authors' hypothesis and potential outcome. This will be also useful for the general audience in understanding the power of *Drosophila* genetics for addressing cell-autonomous functions.

- Thank you for your comment. To clarify our approach and help aide the audience in the design of our study we have generated a new supplemental data file (Supplemental Data 12). Herein, we provide a clear description of the lines used within each figure and include pertinent information regarding the strains (mutant line, GAL4 driver, UAS- line, etc), our rationale for utilizing each line, and a description of the strengths and potential limitations of each line.

2. Related to my concern 1, the authors frequently use ELAV and GMR drivers however, did not differentiate clock-mediated neuroprotective functions vs photoreceptors/visual system. When using a Pan-neuronal driver like ELAV, how modulation of circadian clock or DR affecting the neuronal system? Based upon their findings, it is clear that the life-span and phototaxis performance will be affected by the para-neuronal ELAV driver. Therefore, most of the outcomes shown by the authors could be neuronal as well. Therefore, it will be essential to differentiate neuronal vs visual systems in the entire manuscript. Also, as shown in Fig.5, it is important to clarify how knock-down of GMR-driven DR-sensitive eye-specific Clk-output genes reduces survival and affects phototaxis performance? Recent studies have shown that GMR-expression is not limited to the visual system therefore, most of these genes knock-down, including reduces survival and affects phototaxis performance might be not limited to the visual system.

- We agree that a major limitation of utilizing the Elav-GS-GAL4 driver to over-express the CLK-Δ1/2 lines is the inability to decipher whether the outcomes on eye-aging and/or lifespan are confounded by potential changes in extra-ocular neuronal subtypes. To directly test the role of photoreceptor clocks in diet-mediated changes in visual function and lifespan, we crossed a photoreceptor-specific GAL4 driver (trpl-GAL4;GAL80) with the UAS-CLK-Δ1 line (prCLK-Δ1). Additionally, we crossed the Trpl-GAL4;GAL80 line with a UAS-Clk line (prCLK-OE) to test the effects of over-expressing wild-type *Clk* specifically in photoreceptors. *Note: The Trpl-GAL4;GAL80 line is temperature sensitive and allows for a repression of GAL4 throughout development by housing the flies at 18°C. Once the flies are sorted onto the food they are moved to 30°C to de-repress the GAL4 for the remainder of their lifespan. To avoid potential confusion between the pan-neuronal Elav-GS-GAL4 (nCLK-Δ1) and photoreceptor specific lines (prCLK-Δ1 and prCLK-OE), we placed the eye-specific Clk data in its own figure (Fig. 3). Within Figure 3, we now show that prCLK-Δ1 display accelerated declines in positive phototaxis (3a), reduced ERG amplitude (3b), and massive photoreceptor degeneration by day 10 (3c). Inversely, prCLK-OE improved positive phototaxis (3a) and increased ERG amplitude(3b) in comparison to control flies. Additionally, we found that prCLK-Δ1 flies displayed significant reductions in median lifespan on both AL and DR, as compared to controls (Fig. 4f). This indicates that reducing CLK function within photoreceptor cells is sufficient to decrease lifespan.

*Fig. 3a-c

*Fig. 4f

- We thank the reviewers for their concern per utilizing the GMR-GAL4 driver. Our rationale for using GMR-gal4 for our knockdown experiments with *retinin*, *Gbeta76c*, and *Sunglasses* was the strength of the driver. Namely, we sought to utilize a strong GAL4 driver to promote efficient RNAi-mediated knockdown of these genes within the eye. Additionally, we chose to study *retinin*, *Gbeta76c*, and *sunglasses* based on our bioinformatic analyses demonstrating them to be photoreceptor enriched genes, and because they have clearly defined functional roles within the *Drosophila* eye literature. Therefore, it is our belief that the phototaxis and lifespan changes observed with GMR-GAL4 crossed to *retinin*-RNAi, *Gbeta76c*-RNAi, and *sunglasses*-RNAi is primarily due to their knockdown within the eye.

3. It is well known that GMR-driver mediated expression/knock-down directly affects ommatidia organization/degeneration. Did the authors see any defects in ommatidia organization/degeneration with the ocular expression of CLK-delta1 and CLK- delta or ocular knock-down of ATP or other genes shown in listed in Fig. 5? It will be also important to know if this was impacted by DR or other genetic modulations.

- We agree and are aware of previous studies demonstrating that the GMR-GAL4 driver can induce photoreceptor degeneration due to toxicity from excessive amounts of intracellular GAL4 protein. Although we cannot rule out the possibility that GAL4 toxicity could influence the phenotypes we reported, it is our belief that our findings and conclusions were made with the proper controls. For each of our GMR-GAL4 experiments, we used a control RNAi group: GMR-GAL4>UAS-RNAi control (UAS-mCherry-RNAi, from BDSC) and GMR-GAL4>UAS-RNAi control (UAS-empty vector, from VDRC). The F1 generations from the GMR-GAL4 crosses are GMR heterozygotes, as are the experimental groups. Therefore, it is our assumption that all GMR-GAL4 F1 generations within our experimental RNAi groups (*retinin*, *Gbeta76c*, *ATPalpha*, etc) express a similar amount of GAL4 protein as compared to the RNAi control flies, and thus it is unlikely that GAL4 toxicity differs among our experimental and control groups. Furthermore, we chose our RNAi controls because they share the same genetic background as our experimental UAS- lines. Although others have demonstrated GAL4 toxicity, we do not believe this is a major concern given that our GMR-GAL4>RNAi control groups display normal phototaxis behavior and lifespan compared to wildtype lines (e.g., *Canton-S* etc.).
- When we crossed the GMR-GAL4 to UAS-CLK-Δ1 or UAS-CLK-Δ2, we found that these flies develop to pupae but never eclose. This is likely caused by developmental toxicity from expressing dominant negative CLK. We observed a similar result with both the Trpl-GAL4 (photoreceptor driver, *without temperature-sensitive Gal80) and Spa-GAL4 (a cone-cell driver) when they were crossed to the UAS-CLK-Δ1/2 lines. Therefore, we combined the Trpl-GAL4 driver with the temperature-sensitive GAL80 line to make a photoreceptor-specific, temperature controlled GAL4 driver: The prCLK-Δ1 and prCLK-OE flies allowed us to assess how loss or gain of CLK function (post-developmentally) influenced photoreceptor function and degeneration (as described above).
- Given that *ATPalpha* is highly expressed in cone-cells in addition to photoreceptor cells, we also knocked down *ATPalpha* with the cone-cell specific driver Spa-GAL4. We found that these flies also have reduced phototaxis and lifespan, albeit to a lesser extent compared to GMR-GAL4>*ATPalpha*-RNAi (Supplemental Fig. 8a-b). These data are in support of our initial claim that forcing photoreceptor degeneration by altering ATPalpha levels is sufficient to shorten lifespan. We do not have the ability to tease out whether the difference in lifespan with *ATPalpha* knockdown in GMR-GAL4 vs Spa-GAL4 is due to differences in tissue expression patterns (knockdown in all eye cells vs cone-cells), GAL4 toxicity, knockdown strength, or developmental expression patterns.

*Supplemental Fig. 8a-b

4. As shown in Fig. 3, NCLK-1 flies display elevated immune responses and shortened lifespan and demonstrated that disrupting neuronal CLK function elevates systemic immune responses. Later the authors have demonstrated that this is also eye-driven (Fig. 3c, d). For these experiments, the authors have used fly bodies, instead of heads. Why expression was not tested in the head/eyes?

- We initially observed and reported that there was a significant enrichment in immune markers in the nCLK-Δ1 RNA-Seq. which was performed exclusively within heads (Fig. 4a). Additionally, when we analyzed a publicly available microarray dataset comparing gene expression in the heads of wildtype flies (*y,w*) reared in different lighting conditions, we also observed a significant elevation of immune markers within the heads of flies reared in 12:12hr light dark as compared to constant dark conditions (Supp. Fig. 5f). Taken together,

these results led us to question whether the photoreceptor itself could influence systemic inflammation in the fly. Given that the mammalian literature has extensively documented that retinal degeneration can strongly induce local inflammatory responses in the eye, our main objective with these experiments was to ask whether manipulating photoreceptor homeostasis in the fly could influence systemic inflammation (in the bodies); This is the main reason we decided to quantify AMP expression within bodies while only manipulating processes within the eye/neurons. We found that forcing photoreceptor degeneration (*ATPalpha* knockdown, **Supp. Fig. 5d**) promoted AMP gene expression, while reducing phototransduction (Rhodopsin null lines, **Fig. 4d**) reduced AMP gene expression to varying degrees.

*Fig. 4a and Supplemental Fig. 5f

*Supplemental Fig. 5d and Fig. 4d

5. Similarly, in Fig 5e, why the experiment is limited to the whole fly? Relative mRNA expression of immune genes (*AttaA*, *DiptB*, and *Dro*) calculated by RT-qPCR with mRNA isolated from bodies of *w¹¹¹⁸* and rhodopsin mutant flies housed in 12:12h LD.

- Please see previous response to 4.

6. As I mentioned before, different drivers used for different experiments, however, the conclusion is limited to eye aging? The authors need to justify how the expression of different tissues leads to eyes aging? A clear cell-autonomous vs non-cell-autonomous justification will be required including editing Fig. 6. If neuronal damage signals propagate throughout the body to drive systemic immune responses, the authors need to find basal expression of these genes responsible for immune response in the neuronal and visual tissue/system.

- As mentioned above, the inclusion of the prCLK-Δ1 and prCLK-OE data now demonstrate a clear cell-intrinsic role for CLK in modulating photoreceptor (eye) aging.
- To limit any potential confusion or overstatement of our findings, we have removed the “Systemic Inflammation” from the summary figure (now **Fig. 7**) and have made changes to the discussion.

*Fig. 7

7. Again, the authors have argued that neuronal CLK function is required for the full lifespan extension mediated by DR and indicate that photoreceptor clocks are essential for the maintenance of visual function with age and organismal survival? I think these are confusing statements and need more clarity and justification. Despite several pieces of evidence provided by authors for the neurological role, I am not sure how the outcome is limited to visual function/system?

- We thank the reviewers for their concern. We believe the addition of the new prCLK-DN1 and prCLK-OE data indicate a clear role for CLK function in photoreceptor function and degeneration with age (Fig. 3 and Supplemental Fig. 4). Furthermore, the lifespan shortening observed in prCLK-Δ1 flies (Fig. 4f) indicates that loss of CLK function only within photoreceptors (and post-developmental) is sufficient to reduce lifespan.

8. DR-protection against lifespan shortening downstream of light and/or rhodopsin-mediated signaling in a manner that requires light-adaptation, and by extension, arr1-mediated rhodopsin endocytosis. This justification needs more clarification.

- We apologize for any confusion as to our rationale for analyzing phototaxis responses and lifespan with knockdown of *arr1*. We have added the following sentence within the main text to help clarify: “*arr1* mRNA is circadian in wildtype heads (Supplemental Fig. 1m) and is also a direct CLK target (CLK ChIP-Seq., Supplemental Table 1). Additionally, *arr1* expression was significantly downregulated in heads of nCLK-DN1 flies compared to controls (Supplemental Fig. 7c).” *Note, we have generated Supplemental Fig. 7c for this revision.

C

*Supplemental Fig. 1m and 7c

- Additionally, since we ourselves have not quantitatively measured *Arr1*-mediated rhodopsin endocytosis in flies reared on either AL or DR we agree that our initial statement may have overstated our findings. Therefore, we decided to replace the concluding sentence that is in question with: “Together, these data suggest that modulation of rhodopsin-mediated signaling is sufficient to regulate lifespan in *Drosophila*.”

9. Despite the noticeable difference of immune system and photoreceptors cells between, *Drosophila* with the mammalian visual system, the authors did not indicate the limitations of their finding.

- As mentioned above, we have listed specific limitations of the individual lines used within this study in Supplemental Data 12. Although, we have attempted to highlight potential similarities and differences between our findings and those previously published in the mammalian literature in our Supplemental Discussion 1 section, we do not believe that our utilization of *Drosophila* is in any way a limitation of our study and/or findings.

10. The discussion section of the main text needs a significant improvement including clarity about the above-mentioned comments. The authors have also used discussion/significance of their findings in the SI section without any clarification.

- We have re-worded portions of our main text to improve clarity. The additional discussion text was included within the supplemental section in accordance with Nature Communications guidelines and to adhere to the sizing/formatting of the manuscript.

REVIEWER COMMENTS

Reviewer #1 (Remarks to the Author):

The authors have significantly improved the manuscript. However, I believe that there is still a major problem with the conclusion that the CLK-DN effect that is reported here is related with the circadian clock, as detailed below.

p5-6 and Sup Fig 1

The authors indicate that cycling genes experiment made in light-dark (LD) conditions are circadianly-regulated. I am afraid that there is some misunderstanding here. In LD cycles, cycling genes might be either circadianly-regulated genes or light-regulated genes. I thus do not understand why the term circadian is used here. A DD experiment would be required to show that circadian control is involved here.

This is particularly relevant for the *tim0* mutants where no circadian transcription is known to occur, in contrast to light-regulated transcription (see Wijnen et al., PLoS Genet 2006). Thus, finding 480-490 genes cycling in LD conditions does not mean that there are under circadian regulation. Having a circadian transcription in *tim0* flies would be highly surprising and goes against all previous work, to my knowledge.

This is a key point that needs to be clarified since the effect of DR could well be on light-regulated transcription more than on circadianly-regulated transcription.

Figure 1 and sup fig3e

The above comment brings back the question of the CLK-DN effect. I understand that the RU486 conditional expression system that is use here and that is confirmed by the TS-gal80 system supports a non-developmental effect, and the use of the *clkout* mutant is also a good control for non-specific CLK-DN effects. However, it is known that CLK affects the transcription of non-circadianly controlled genes (McDonald et al., Cell 2001, Abruzzi et al., Genes Dev 2011), possibly including light-controlled genes. I would thus suggest to interpret the data differently, with DR affecting reinforcing light-regulated transcription and not circadian transcription.

In the rebuttal letter, the authors indicate that because TIM/PER repress CLK-dependent transcription, the corresponding mutants cannot be used here to confirm the circadian function of *clk*. I disagree with this, constitutive *clk* expression in the *per0* and *tim0* mutants does not change the fact that the clock function is abolished. I still would like to see a phototaxis experiment with *tim0* or *per0* mutants to really make the difference between a function of the CLK transcription factor and the circadian clock.

Reviewer #2 (Remarks to the Author):

the authors have addressed my concerns. As such I support the publication of the manuscript.

Reviewer #3 (Remarks to the Author):

The authors did a really good job in addressing most of the concerns/questions I raised in the previous version. In addition to addressing questions, the authors have provided additional supporting data for their findings. However, a couple of important questions still need some clarification/justification with the data support. Therefore, addressing the following questions will make this study suitable for publications in the prestigious Nature Communications Journal.

1. The authors have indicated that the GMR-GAL4 driver can induce photoreceptor degeneration due to toxicity from excessive amounts of intracellular GAL4 protein. This justification is very poor. It is well established for several neurological disease models that GMR-Gal4 specific expression of mutant, not wild-type associated with ommatidia organization/degeneration. Instead of going through this intracellular toxicity theory, the authors should directly if they have received any ommatidia organization/degeneration or not?

2. The authors mentioned that when they crossed the GMR-GAL4 to UAS-CLK-delta1 or UAS-CLK- delta 2, they found that these flies develop to pupae but never eclose. How expression of UAS-CLK- delta1 or UAS-CLK- delta2 with GMR-Gal4 driver caused pupae stage lethality. It is hard to believe ocular-specific expression leads to developmental toxicity from expressing dominant-negative CLK. In my opinion, the lethality is caused because GMR-driver is not specific to ocular and is shown to express in other tissues as well. Did the authors check the expression of GMR-GAL4 in other tissues as indicated in the other tissues? <https://pubmed.ncbi.nlm.nih.gov/22911584/>
<https://pubmed.ncbi.nlm.nih.gov/26440079/>
Related to this the authors have indicated that they have similar results with both the Trpl-GAL4 (photoreceptor driver, *without temperature-sensitive Gal80). Therefore, the authors need to explain how photoreceptor-specific expression of UAS-CLK-delta1 or UAS-CLK- delta 2 caused developmental lethality?

3. It is still hard to understand how eye-specific expression of leads to elevates systemic immune responses beyond the head as in the updated Fig.7, the authors did not explain that (including elevated AMP expression within bodies). They still need to justify the elevated level of immune response beyond ocular tissue.

4. In general, the author needs to say that some of the lethality/immune response could be due to the expression of UAS-CLK-delta1 or UAS-CLK- delta 2 expression of these dominant negative in other tissues, beyond their ocular (eye)-specific expression.

Reviewer #1 (Remarks to the Author):

The authors have significantly improved the manuscript. However, I believe that there is still a major problem with the conclusion that the CLK-DN effect that is reported here is related with the circadian clock, as detailed below.

p5-6 and Sup Fig 1

The authors indicate that cycling genes experiment made in light-dark (LD) conditions are circadianly-regulated. I am afraid that there is some misunderstanding here. In LD cycles, cycling genes might be either circadianly-regulated genes or light-regulated genes. I thus do not understand why the term circadian is used here. A DD experiment would be required to show that circadian control is involved here.

This is particularly relevant for the *tim0* mutants where no circadian transcription is known to occur, in contrast to light-regulated transcription (see Wijnen et al., PLoS Genet 2006). Thus, finding 480-490 genes cycling in LD conditions does not mean that there are under circadian regulation. Having a circadian transcription in *tim0* flies would be highly surprising and goes against all previous work, to my knowledge.

This is a key point that needs to be clarified since the effect of DR could well be on light-regulated transcription more than on circadianly-regulated transcription.

We thank the reviewer for the comments to improve our manuscript. In our previous submission we reported that roughly 480-490 transcripts oscillate in a circadian fashion in *tim⁰¹* flies. We did not intend to argue, or make the distinction, that the oscillating genes we reported for the *tim⁰¹* flies were driven by “circadian” vs “light-driven” mechanisms. Our only intention was to state the number of transcripts that displayed a circadian expression pattern on AL or DR according to our JTK_CYCLE parameters.

We appreciate the reviewer’s concern that the oscillating transcripts reported for *CantonS* wildtype flies may be influenced by light-dark cycles, as we did not perform additional time-course microarrays in free-running conditions (DD). Our laboratory has a long history of characterizing DR-mediated lifespan benefits while housing flies in LD, and we designed our time-course microarray experiments to identify the circadian processes that are enriched on AL and DR diets in this context. It is well established that *tim⁰¹* mutant flies are behaviorally arrhythmic in constant conditions (DD) but retain some ability to entrain to light-dark cycles due to masking effects. It is our assumption that any transcripts that display a circadian expression pattern in the *tim⁰¹* flies housed in 12:12h LD are most likely “light-driven”.

To classify which genes are circadianly-regulated in wildtype flies we subtracted the light-driven transcripts (i.e., also circadian in *tim⁰¹*) from the transcripts that comprise the AL and DR circadian microarrays in wildtype *CantonS* flies. By controlling for the potential confounding variable of light, we believe that the remaining genes (circadian in *CantonS* but not in *tim⁰¹*) are clock controlled. However, it is possible that these circadianly-regulated transcripts are also reinforced by lighting signals/cues. For instance, light-mediated intracellular signaling cascades (e.g., photoentrainment by *Cry*) may impinge on the circadian transcription factor CLK to influence timing (i.e., phase shifts) and/or amounts of transcriptional activation of downstream targets (i.e., amplitude). Furthermore, light- and nutrient-signals from the diet may independently or synergistically converge on the core-clock machinery to influence the transcription of clock-controlled genes/processes within photoreceptors.

We have provided the following statements to clarify these concerns within the text:

*“In LD conditions, circadian transcripts may be driven by the core-molecular clock or by rhythmic lighting cues. We performed time-course microarrays under similar conditions in arrhythmic *tim*⁰¹ mutant flies which lack circadian transcriptional rhythms, as we reasoned that the circadian transcripts identified in these mutants are primarily light-driven. Circadianly-regulated transcripts (clock-output genes) were identified as those which oscillate only in wildtype flies and not in *tim*⁰¹ mutants (Supplementary Fig. 1a).”*

The following figures and datasets have been updated to reflect our new analyses:
Fig. 1a-f, Supplementary Fig. 1a-b, f-g, j-k and Supplementary Data 1.

Figure 1 and sup fig3e

The above comment brings back the question of the CLK-DN effect. I understand that the RU486 conditional expression system that is use here and that is confirmed by the TS-gal80 system supports a non-developmental effect, and the use of the clkout mutant is also a good control for non-specific CLK-DN effects. However, it is known that CLK affects the transcription of non-circadianly controlled genes (McDonald et al., Cell 2001, Abruzzi et al., Genes Dev 2011), possibly including light-controlled genes. I would thus suggest to interpret the data differently, with DR reinforcing light-regulated transcription and not circadian transcription.

We agree with the reviewer on being cautious in the interpretation of our experiments and have modified the text to reflect this. In any experiment where CLK function is perturbed, there will always be the possibility that the phenotypic observations reported are due to changes in the expression of downstream circadian genes, non-circadian genes, or a combination of both gene sets. Regarding our analyses of CLK's role in mediating the beneficial effects of DR on photoreceptor physiology with age, we believe that the effects are more-likely due to changes in circadian vs non-circadian gene expression. Our initial analyses of the AL/DR circadian transcriptomes led us to identify many light and eye-related circadian transcripts. We found that the circadian phototransduction genes displayed increased expression and circadian amplitude in *CantonS* flies on DR but were no longer rhythmic in *tim*⁰¹ flies, suggesting that their circadian expression pattern in wildtype flies is regulated downstream of the molecular clock as opposed to light.

This argument was strengthened by our nCLK-Δ1 RNA-Seq. experiments: We found that genes that were significantly downregulated in expression in nCLK-Δ1 heads (DR) and were also circadian in wildtype flies displayed a significant enrichment for biological processes related to light-sensing and homeostatic processes within the eye (Supplemental Fig. 2c). Genes that were significantly downregulated in expression in nCLK-Δ1 heads but were non-circadian in wildtype flies failed to display a significant enrichment for light-responsive pathways or eye homeostasis. Below is a table comparing the gene-ontology enrichment scores for genes that were significantly down-regulated in nCLK-Δ1 (on DR) and are either circadian (“Circadian_Enrichment”) or non-circadian (“Non-Circadian_Enrichment”) in wildtype flies. Therefore, it is our interpretation that DR and CLK primarily influence eye aging via downstream regulation of circadian genes, and not non-circadian genes.

TermID	Term	Circadian_Enrichment	Non-Circadian_Enrichment
GO:0009416	response to light stimulus	3.32E-06	0.7968651
GO:0009642	response to light intensity	1.27E-05	0.5084995
GO:0071482	cellular response to light stimulus	2.60E-05	0.91390436
GO:0016059	deactivation of rhodopsin mediated signaling	2.64E-05	0.71738834
GO:0022400	regulation of rhodopsin mediated signaling pathway	2.64E-05	0.71738834
GO:0007602	phototransduction	5.98E-05	0.43081234
GO:0009583	detection of light stimulus	8.65E-05	0.46534368
GO:0007603	phototransduction, visible light	0.000111058	0.79408137
GO:0006874	cellular calcium ion homeostasis	0.000561518	0.13304433
GO:0009584	detection of visible light	0.000996215	0.65752068
GO:0050953	sensory perception of light stimulus	0.001365551	0.48052537
GO:0001895	retina homeostasis	0.0015157	1
GO:0016056	rhodopsin mediated signaling pathway	0.0015157	0.64171047
GO:0002032	desensitization of G protein-coupled receptor signaling pathway by arrestin	0.001704044	1
GO:0050962	detection of light stimulus involved in sensory perception	0.002159627	1
GO:0007601	visual perception	0.003498833	0.34515896
GO:0071484	cellular response to light intensity	0.004972445	1
GO:0045494	photoreceptor cell maintenance	0.009003586	1
GO:0016062	adaptation of rhodopsin mediated signaling	0.009674064	1
GO:0036367	light adaption	0.009674064	1
GO:0050908	detection of light stimulus involved in visual perception	0.009674064	1
GO:0046154	rhodopsin metabolic process	0.011643387	1
GO:0042052	rhabdomere development	0.013623166	0.75601232

It is possible that DR reinforces CLK-mediated output by influencing light-sensing. Alternatively, light-mediated signals may influence DR's ability to regulate CLK-mediated transcriptional output. In other words, DR may promote CLK-mediated circadian transcription by reinforcing light-mediated signals.

We have updated Supplemental Data 4 has to include the gene-ontology enrichment scores for the genes that are non-circadian in wildtype flies and down-regulated in nCLK- Δ 1 RNA-Seq. (Supplemental Data 4c). We have denoted the changes in the text:

*“Additionally, genes that were both circadian in wild-type heads and downregulated in nCLK- Δ 1 were highly enriched for homeostatic processes related to eye function, while downregulated genes in nCLK- Δ 1 that were non-circadian in wildtype heads displayed no such enrichment (**Supplementary Fig. 2c and Supplemental Data 4**).”* And, in the discussion: *“Given that CLK transcriptionally regulates circadian and*

non-circadian transcripts, future studies may determine whether the time-of-day regulation of these genes by CLK is germane to promoting eye health with age.”

In the rebuttal letter, the authors indicate that because TIM/PER repress CLK-dependent transcription, the corresponding mutants cannot be used here to confirm the circadian function of clk. I disagree with this, constitutive clk expression in the per0 and tim0 mutants does not change the fact that the clock function is abolished. I still would like to see a phototaxis experiment with tim0 or per0 mutants to really make the difference between a function of the CLK transcription factor and the circadian clock.

We have edited our conclusions according to the comments made by the reviewer and no longer state that circadian regulation is required for the DR- or CLK-mediated effects within our study. Thus, the additional *per⁰¹/tim⁰¹* phototaxis experiments are not required to support our conclusions or interpretations.

Reviewer #2 (Remarks to the Author):

the authors have addressed my concerns. As such I support the publication of the manuscript.

Reviewer #3 (Remarks to the Author):

The authors did a really good job in addressing most of the concerns/questions I raised in the previous version. In addition to addressing questions, the authors have provided additional supporting data for their findings. However, a couple of important questions still need some clarification/justification with the data support. Therefore, addressing the following questions will make this study suitable for publications in the prestigious Nature Communications Journal.

1. The authors have indicated that the GMR-GAL4 driver can induce photoreceptor degeneration due to toxicity from excessive amounts of intracellular GAL4 protein. This justification is very poor. It is well established for several neurological disease models that GMR-Gal4 specific expression of mutant, not wild-type associated with ommatidia organization/degeneration. Instead of going through this intracellular toxicity theory, the authors should directly if they have received any ommatidia organization/degeneration or not?

We did not observe overt changes to ommatidia organization/degeneration in any of our GMR-GAL4 crosses utilized in this manuscript. Below we have crossed GMR-GAL4 with UAS-mCD8-GFP which express green fluorescent protein on the cell surface and did not observe any disorganization or degeneration of the ommatidia.

2. The authors mentioned that when they crossed the GMR-GAL4 to UAS-CLK-delta1 or UAS-CLK- delta 2, they found that these flies develop to pupae but never eclose. How expression of UAS-CLK- delta1 or UAS-CLK- delta2 with GMR-Gal4 driver caused pupae stage lethality. It is hard to believe ocular-specific expression leads to developmental toxicity from expressing dominant-negative CLK. In my opinion, the lethality is caused because GMR-driver is not specific to ocular and is shown to express in other tissues as well. Did the authors check the expression of GMR-GAL4 in other tissues as indicated in the other tissues? <https://pubmed.ncbi.nlm.nih.gov/22911584/> <https://pubmed.ncbi.nlm.nih.gov/26440079/>

Related to this the authors have indicated that they have similar results with both the Trpl-GAL4 (photoreceptor driver, *without temperature-sensitive Gal80). Therefore, the authors need to explain how photoreceptor-specific expression of UAS-CLK-delta1 or UAS-CLK- delta 2 caused developmental lethality?

We agree with the reviewer's comment that the observed developmental lethality in crossing the GMR-GAL4 to either UAS-CLK- Δ 1 or UAS-CLK- Δ 2 is surprising and may be due to mis-expression of the GAL4 in non-ocular tissues. Since others have extensively characterized the mis-expression of GMR-GAL4 in non-ocular tissues we did not feel it was necessary for us to recapitulate those studies here, especially as we were able to utilize the Trpl-GAL80-GAL4 line which allowed us to examine the effects of modulating CLK specifically in adult photoreceptors. We did however observe a similar pupae stage lethality when crossing the UAS-CLK- Δ 1 or UAS-CLK- Δ 2 with the SPA-GAL4 driver (a commonly used cone-cell specific driver) and the RDGA-GAL4 driver (an additional eye driver). We do note that it is possible that the developmental lethality observed with SPA or RDGA driver lines may also be caused by mis-expression in extra-ocular tissues.

We would like to clarify our previous response where we stated that the non-temperature-sensitive photoreceptor driver, Trpl-GAL4, flies were 'similar' to the GMR-GAL4 flies when crossed to UAS- CLK- Δ 1 or -CLK- Δ 2, as this statement may have been unintentionally misleading. To be clear, we did not observe any developmental lethality when crossing Trpl-GAL4 flies with either UAS-CLK- Δ 1 or UAS-CLK- Δ 2; these flies eclosed normally. However, we found that upon eclosion, Trpl-GAL4>UAS-CLK- Δ 1 or UAS- CLK- Δ 2 flies displayed very low positive phototaxis responses, and overt disruption to their photoreceptors, measured by transverse sections of the eye (below). These observations suggest that CLK is required for proper development of photoreceptors. Furthermore, these findings were central in our reasoning for using the temperature sensitive Trpl;GAL80-GAL4 line for further analysis.

UAS-CLK-Δ1/Δ2 crossed with eye-specific GAL4 drivers

GAL4 driver	Phenotype
GMR(12)-GAL4	no eclosion
RDGA-GAL4	no eclosion
Spa-GAL4	no eclosion
Trpl-GAL4	extremely low phototaxis and no photoreceptors at eclosion

Figure 1: *trpl-gal4>Clk-DN1* cross-section at day 2.

3. It is still hard to understand how eye-specific expression of leads to elevates systemic immune responses beyond the head as in the updated Fig.7, the authors did not explain that (including elevated AMP expression within bodies). They still need to justify the elevated level of immune response beyond ocular tissue.

We thank the reviewer for their comments to improve our study. Our main reasoning for assessing systemic immune responses and providing those data within this manuscript were due to the fact that we observed large elevations in anti-microbial peptide (AMP) expression when manipulating CLK (nCLK-Δ1) and when forcing photoreceptor degeneration (GMR-GAL4>UAS-ATPalpha-RNAi). AMP expression has been shown to result from local damage responses, but they are primarily upregulated within the fat-body. We do not fully understand the mechanisms by which the *Drosophila* eye and more specifically, the photoreceptors influence systemic immune responses. However, we can speculate on several potential explanations, including: 1. A retinal-blood barrier exists to keep the hemolymph separate from the ommatidia such that the extracellular space surrounding the photoreceptors can maintain a proper ion balance required for photoreceptor viability and the ion gradient required for phototransduction to occur. With age and neurodegeneration, there is leakiness of both the retinal/brain and blood/brain barrier in *Drosophila*. Interestingly, recent reports have indicated that the blood-brain barrier is under direct circadian control. The upregulation of the AMPs in the body, which is indicative of a systemic immune response, may indicate that photoreceptor degeneration can lead to a

breakdown of the retinal brain barrier and damage signals propagating throughout the hemolymph may then activate the fat body to increase the expression of AMPs. 2. An additional explanation may be that as photoreceptors degenerate they do so in a necrotic fashion vs apoptosis. Previous reports have indicated that promoting photoreceptor degeneration by over-expressing a leaky channel can drive local neuronal death in other wise healthy cells. As the photoreceptors degenerate due to age and Ca²⁺ induced toxicity from light exposure, they may promote a feed-forward cycle of neuronal cell death and inflammation that ultimately leads to systemic inflammatory responses.

Since we have only demonstrated the correlations between photoreceptor degeneration and systemic inflammation with age and have not directly claimed that these systemic immune responses are causal for the phenotypes we described (visual function, lifespan). We have added some of these explanation in the revised submission in the discussion

We have provided the following text within the discussion to address these concerns:

“Among the more interesting and unexpected observations of this study is that the Drosophila eye can influence systemic immune responses, as we observed elevated AMP expression in the bodies of flies over-expressing CLK-Δ1 pan-neuronally and in flies with forced photoreceptor degeneration (ATPalpha-RNAi). It is possible that GAL4 mis-expression may promote inflammatory responses in the fly bodies, although we found a reduction in systemic inflammation in the rhodopsin-null lines indicating that this phenomenon can originate at the photoreceptor. We also found that these systemic immune responses correlate with lifespan changes (increased body AMP expression is associated with declines in longevity, and vice-versa), similar to what is reported with chronic inflammation or “inflammaging” in other models. However, we cannot conclude whether neuronal or eye-mediated increases in systemic inflammation are at all causal to aging in other tissues. Furthermore, we do not fully understand the mechanisms by which the Drosophila eye and more specifically, the photoreceptor influences systemic immune responses although we speculate that photoreceptor degeneration may disrupt the retinal-blood barrier such that damage signals from the eye may propagate through the hemolymph to activate AMP expression in distal tissues. Future studies are aimed at elucidating the mechanisms by which the eye influences systemic inflammation and its relation to aging and longevity.”

4. In general, the author needs to say that some of the lethality/immune response could be due to the expression of UAS-CLK-delta1 or UAS-CLK- delta 2 expression of these dominant negative in other tissues, beyond their ocular (eye)-specific expression.

To address this potential limitation, we have provided the follow text within the manuscript:

“However, as with all tissue-specific driver systems, we cannot rule of the possibility that our ELAV-GS-GAL4 driver expresses in a small population of non-neuronal cell-types, which, in theory, could contribute to the elevated systemic inflammatory responses and/or influence lifespan.”

REVIEWERS' COMMENTS

Reviewer #1 (Remarks to the Author):

The changes made in the new revised version of the manuscript addresses my concerns.

I have one last request about the circadian versus non-circadian 24h oscillations:

The authors now define circadian transcripts as LD cycling transcripts that do not cycle in tim0 clockless flies and I think that it is a very reasonable estimation for a circadian control.

In the modified sentence “In LD conditions, circadian transcripts may be driven by the core-molecular clock or by rhythmic lighting cues...”, please do not use circadian for cycling transcripts in LD. I would suggest “In LD conditions, transcripts showing 24h oscillations...”

This should be applied to the other instances of the word circadian in the text, please keep circadian when speaking about clock-dependent expression otherwise it is confusing for the reader.

Reviewer #3 (Remarks to the Author):

All the previous concerns have been appropriately addressed and this should be now suitable for publication in Nature Communications. Best wishes

REVIEWERS' COMMENTS

Reviewer #1 (Remarks to the Author):

The changes made in the new revised version of the manuscript addresses my concerns. I have one last request about the circadian versus non-circadian 24h oscillations:

The authors now define circadian transcripts as LD cycling transcripts that do not cycle in *tim0* clockless flies and I think that it is a very reasonable estimation for a circadian control.

In the modified sentence "In LD conditions, circadian transcripts may be driven by the core-molecular clock or by rhythmic lighting cues...", please do not use circadian for cycling transcripts in LD. I would suggest "In LD conditions, transcripts showing 24h oscillations..."

This should be applied to the other instances of the word circadian in the text, please keep circadian when speaking about clock-dependent expression otherwise it is confusing for the reader.

We thank the reviewer for their comment and have made the appropriate changes to the text: "In LD conditions, transcripts displaying 24h oscillations may be driven by the core-molecular clock ("circadian") or by rhythmic lighting cues [13]."

Reviewer #3 (Remarks to the Author):

All the previous concerns have been appropriately addressed and this should be now suitable for publication in Nature Communications. Best wishes

We appreciate the reviewers' comments and suggestions throughout this review process.